



# UK surface NO₂ levels dropped by 42% during the COVID-19 lockdown: impact on surface O₃

James D. Lee[1], Will S. Drysdale[1], Doug P. Finch[2], Shona E. Wilde[1] and Paul I. Palmer[2].

[1]National Centre for Atmospheric Science, Department of Chemistry, University of York, York, UK
5 [2]School of GeoSciences, University of Edinburgh, Edinburgh, UK.

*Correspondence to: James Lee (james.lee@york.ac.uk)*

## Abstract

10 We report changes in surface nitrogen dioxide ($NO_2$) across the UK during the COVID-19 pandemic when large and rapid emission reductions accompanied a nationwide lockdown (23rd March—31st May, 2020, inclusively), and compare them with values from an equivalent period over the previous five years. Data are from the Automatic Urban and Rural Network (AURN) that form the basis of checking nationwide compliance with ambient air quality directives. We calculate that $NO_2$ reduced by 42% on average across all 126 urban AURN sites, with a 15 slightly larger (48%) reduction at sites close to the roadside (urban traffic). We also find that ozone ($O_3$) increased by 11% on average across the urban background network during the lockdown period. Total oxidant levels ($O_x = NO_2 + O_3$) increased only slightly on average (3%), suggesting the majority of this change can be attributed to photochemical repartitioning due to the reduction in $NO_x$. Generally, we find larger, positive $O_x$ changes in southern UK cities which we attribute to increased UV radiation and temperature in 2020 compared to previous 20 years. The net effect of the $NO_2$ and $O_3$ changes is a sharp decrease in exceedances of the $NO_2$ air quality objective limit for the UK, with only one exceedance in London in 2020 up until the end of May. Concurrent increases in $O_3$ exceedances in London emphasize the potential for $O_3$ to become an air pollutant of concern as $NO_x$ emissions are reduced in the next 10-20 years.

## 1 Introduction

The current Coronavirus SARS-CoV-2 (COVID-19) outbreak was first identified in Wuhan, China, in December 2019, and was recognised as a pandemic by the World Health Organization (WHO) on 11 March 2020 (WHO, 2020). As of early August 2020, there have been almost 18 million confirmed cases and over 700,000 deaths 30 reported across the world (https://coronavirus.jhu.edu/map.html). Efforts to prevent the virus spreading have included severe travel restrictions and the closure of workplaces, inevitably leading to a significant drop in emissions of primary air pollutants from several important sectors. This has provided a unique opportunity to examine how air pollutant concentrations respond to an abrupt and prolonged perturbation, followed by policy-relatable increases as restrictions are incrementally relaxed. Here, we report changes in nitrogen dioxide ($NO_2$) 35 across the UK and discuss them in context of observed changes in surface ozone ($O_3$).

In 2017 the road transport sector accounted for 32 % of UK $NO_x$ (sum of NO and $NO_2$), the largest emission from a single sector, followed by energy industries (21 %), manufacturing industries and construction (17 %), and non-road transport (14 %) (https://www.gov.uk/government/statistical-data-sets/env01-emissions-of-air-pollutants). 40 In major cities, the contribution from transport is typically much higher, e.g. 53% in Greater London (Vaughan et



al., 2016). In recent years, there has been a pronounced reduction in $NO_x$ emission (Defra, 2018a) that largely reflects lower transport emissions. Since 2014, Euro 6 standards for light passenger diesel vehicles reduced the maximum permitted $NO_x$ emission from 0.18 to 0.08 g/km, and the number of ultra-low emission vehicles (e.g. electric, hybrid cars) has increased its market share from 0.59% in 2014 to achieved 2.6% in 2018. Despite these
developments, air pollution is still currently the largest environmental health stressor on the UK population (Public Health England, 2019).

At present the main pollutants of concern in urban centres are $NO_2$ and particulate matter with radii smaller than 2.5 microns ($PM_{2.5}$), and in suburban and rural environments is $O_3$, with exposure to excess levels of these species
is known to have a negative effect on human health (An et al., 2018; Kurt et al., 2016; Mannucci et al., 2015). $O_3$ is a secondary air pollutant formed photochemically by the oxidation of volatile organic compounds (VOCs) in the presence of $NO_x$ (Monks et al., 2015). It is generally lower in urban areas due to reactions with $NO_x$, but in the past two decades over the UK (Finch and Palmer, 2020), and across the world (Fleming et al., 2018; Lefohn et al., 2018; Ma et al., 2016; Paoletti et al., 2014; Sicard et al., 2013; Sun et al., 2016), there have been large mean
surface $O_3$ increases driven by reduced $NO_x$ emissions. Air pollution has led to an estimated 29,000 premature deaths/year in the UK, equivalent to 340,000 life years across the population in any one year and costs the UK economy between £10 billion and £20 billion/year (Royal College of Physicians, 2016). To meet the UK Government's clean air strategy (UK Government, 2019) and its commitment to achieve zero carbon emission target by 2050, sales of non-zero emission cars, vans and motorcycles will end by 2035. One of the challenges
associated with the progressive move to a low-NOx vehicle fleet in the UK is to understand the impacts on surface air pollution if other emissions are not reduced commensurately.

The widespread and rapid reduction in UK transport activity (and therefore the associated emissions) from the COVID-19 lockdown represents a natural experiment to study air pollution with a greatly reduced volume of $NO_x$-
emitting vehicles that we use as a proxy for a future low-$NO_x$ vehicle fleet. Figure 1 summarises the timeline of events associated with COVID-19 in the UK, including Google mobility data that describe the percentage changes in transport from a pre-lockdown baseline and daily mortality values reported by the UK Office of National Statistics. The UK Foreign and Commonwealth Office issued a travel advisory on 28[th] January not to travel to mainland China. The first two UK cases of COVID-19 were confirmed on 31[st] January, with a third case confirmed
on 6[th] February. As the number of cases continues to rise, the first UK death from COVID-19 was confirmed on the 5[th] March.  On that same day, the UK government moved from the "containment" to the "delay" phase of addressing COVID-19, which included, for example, social-distancing. A UK-wide lockdown was announced nearly three weeks later on 23[rd] March, with citizens instructed to stay at home with the exception of shopping for basic necessities and one form of exercise per day, medical needs and travel associated with key workers. An
immediate effect of these restrictions in movement was a large and progressive drop in transport use, with an associated reduction in motor vehicles throughout the lockdown period. We use *in situ* measurements collected across the UK to examine how these reductions (and other changes) have affected $NO_2$ in the UK, with a discussion on how this has in turn affected $O_3$. We also examine the changes in exceedances of limit values for $NO_2$ and $O_3$ and assess whether the COVID-19 lockdown can provide useful information on how air pollution
will respond to future changes in emissions due to the move to a low-carbon economy.  In the next section we





discuss the data we use, in section 3 we describe our results for NO$_2$ that we put into context in section 4 with the observed changes in surface O$_3$, as well as comparing our results with other studies. We conclude the paper in section 5.

## 2 Data and Methods

### 2.1 In situ Measurements of NO$_2$ and O$_3$

We use data collected as part of the Defra Automatic Urban and Rural (AURN) network, currently consisting of 150 active sites across the UK (Figure S1 and Tables S1 and S2) and is the main network used for compliance reporting against the Ambient Air Quality Directives. It includes automatic air quality monitoring stations measuring oxides of nitrogen (NO$_x$), sulphur dioxide (SO$_2$), ozone (O$_3$), carbon monoxide (CO) and particles (PM$_{10}$, PM$_{2.5}$). Online measurements of VOCs are available at a small number of sites. These sites provide hourly information which is communicated rapidly to the public, using a wide range of electronic media and web platforms. More detail can be found at https://uk-air.defra.gov.uk. Three different site types are used in this analysis. Urban traffic sites are defined as being in continuously built-up urban areas, with pollution levels predominantly influenced by emissions from nearby traffic. Urban background sites are located such that pollution level is not influenced significantly by any single source or street, but rather by the integrated contribution from all sources upwind of the stations. These can be considered more representative of residential areas. Rural background sites are sited more than 20 km away from agglomerations and more than 5 km away from other built-up areas, industrial installations or motorways or major roads, so that the air sampled is representative of air quality in a surrounding area of at least 1,000 km$^2$.

The AURN network uses standardised techniques and operating procedures to ensure data are comparable. Full details can be found at https://uk-air.defra.gov.uk/assets/documents/reports/empire/lsoman/ but a brief description will be given here. Nitric oxide (NO) in the sample air stream reacts with O$_3$ in an evacuated chamber to produce activated nitrogen dioxide (NO$_2$*). This then returns to its ground (un-activated) state, emitting a photon (chemiluminescence). The intensity of the chemiluminescent radiation produced depends upon the amount of NO in the sampled air. This is measured using a photomultiplier tube (PMT) or photodiode detector, so the detector output voltage is proportional to the NO concentration. The ambient air sample is divided into two streams. In one stream, the ambient NO$_2$ is reduced to NO (with at least 95% efficiency) using a molybdenum catalyst converter before reaction. The molybdenum converter should be at least 95% efficient at converting NO$_2$ to NO. External gas cylinders or an internal permeation oven and zero air scrubber are used to provide daily automatic check calibrations for NO. The NO$_2$ conversion efficiency is checked every 6 months using either an NO$_2$ calibration cylinder or gas phase titration of the NO with O$_3$. In recent years it has become well established that NO$_2$ measurements using molybdenum converters can overestimate NO$_2$ due to interferences from other oxidised nitrogen species (e.g. HNO$_3$, PAN, HONO) (Steinbacher et al., 2007). However, in urban environments the interferences are often minimal compared to the levels of NO$_x$ (Villena et al., 2012). Ozone is measured by UV absorption at 254nm, with concentrations calculated using the Beer-Lambert Law (Parrish and Fehsenfeld, 2000). An O$_3$-removing scrubber is used to provide a zero-reference intensity. An internal ozone generator and zero air scrubber are used to provide daily automatic check calibrations and instruments are calibrated with a primary ozone standard every 12 months. Whilst the accuracy of the measurement will vary on a site by site basis, the



maximum allowed uncertainty for the AURN network is 15% for $NO_2$ and $O_3$ measurements and 25% and for $PM_{2.5}$. To study the effect of the lockdown on $NO_2$ levels in the UK, we use measurements from 66 Urban Traffic and 62 Urban Background sites across the UK, all that have measurements between 2015 to the end of May 2020.

**2.2 Correlative Meteorological Data**

Measured meteorological data (wind direction, wind speed and temperature) is not available at most AURN sites so modelled data, based on the position of the site, from the UK Met Office unified model is used. UV-A irradiance data is taken from measurements made by the Public Health England (PHE) solar network.

**2.3 Statistical Methods**

To quantify the impact of the COVID-19 lockdown on atmospheric levels of $NO_2$ and $O_3$, we compare measurements during the lockdown with values corresponding to 'business as usual' (BAU), i.e. what we would have expected in the absence of the pandemic. To determine our BAU scenario, we first linearly detrend and de-seasonalise $NO_2$ data at each AURN site based on the climatology of the previous five years (from January $1^{st}$

2015 to December $31^{st}$ 2019). This five-year period is sufficiently long to take into account year to year variations in meteorology but short enough to reduce the impact of any longer-term trends driven by earlier changes in emission standards (e.g. the introduction of Euro VI in 2017 and an ultra-low emission zone in London in 2019). We then calculated the difference between a linear regression model of the previous data, projected forward to June 2020 to predict BAU values of $NO_2$ and $O_3$ (Figure 2) and calculated the difference between this and the

measured values. We acknowledge there are uncertainties associated with our approach, but this method offers simplicity and straightforward error propagation. Other more complex methods to determine BAU that, for example, explicitly take into account local changes in meteorology (Grange and Carslaw, 2019) will also be subject to uncertainties, e.g. the extent which regional-scale meteorological fields can describe smaller-scale variations in atmospheric pollutants. We define the start of the UK lockdown period as the 23th March 2020 when

the lockdown was advised by the UK government. Figure 1 shows that a decrease in mobility in the transport sector is already evident from the $9^{th}$ March, which in the absence of any obvious change in law is perhaps influenced by the emerging crises in nearby European countries. Our analysis concludes on $31^{st}$ May 2020 the day before the first phase of lockdown easing in England.

We use independent sample z-tests to test the significance of changes in mean concentration for each site between the lockdown period and the mean of same period for the past five years, the lockdown period and measurements in 2020 prior to the lockdown and measurements from prior to lockdown in 2020 with the same period for the previous five years. This test indicates how likely the observed changes in mean concentration between the different time periods are due to chance and noise in the data or whether they are statistically significant and be

attributed to a real signal, which in our work is the start of the lockdown.

**3 Results**

Figure 2 shows UK mean deseasonalized $NO_2$ and $O_3$ observations from all urban sites and the mean trend from

2015 to 2019. The mean $NO_2$ linear trend across all AURN urban traffic (background) sites is -1.4 (-0.6) µg m$^{-3}$





yr$^{-1}$ (-4.5 (-2.1) % yr$^{-1}$). The urban traffic site at London Marylebone Road shows the largest decreasing trend over the past five years of -5.5 μg m$^{-3}$ yr$^{-1}$ (-6.7 % yr$^{-1}$), whereas eight urban sites show a small increasing trend in NO$_2$ between 0.1 - 0.6 μg m$^{-3}$ yr$^{-1}$ (0.5 – 1.2 % yr$^{-1}$). The mean O$_3$ linear trend across all urban traffic (background) sites is 2.4 (1.3) μg m$^{-3}$ yr$^{-1}$ (5.5 (3.1) % yr$^{-1}$).


### 3.1 Meteorological Context

It is well understood that ambient concentrations of air pollutants are greatly affected by meteorology, with low wind speeds causing a build-up of pollutants and over the UK easterly flow is often accompanied by pollution from mainland Europe. Figure 3 shows surface wind data from six cities across the UK (London, Bristol, Cardiff, 170 Newcastle, Glasgow and Belfast), providing information from a wide geographical range across the country. Wind roses for the pre (10$^{th}$ January – 10$^{th}$ March) and post (23$^{rd}$ March – 31$^{st}$ May) lockdown periods of 2020 and the mean of 2015-2019 show that all cities during the pre-lockdown period in 2020 were dominated by strong westerly winds across all of the UK, with successive low-pressure systems across the UK including the named storms Ciara, Dennis and Jorge through the month of February and early March. The winter season (January-February) 175 was the fifth wettest on record and the fifth warmest. February 2020 was the wettest ever February recorded in the UK. The wind roses also show that 2020 saw much stronger winds than the mean of the previous five years. Since the beginning of the COVID-19 lockdown, meteorological conditions have been much more settled, with high pressure and easterly winds dominating UK weather since mid-March, especially in southern and western UK. Typically, these meteorological conditions are associated with higher levels of air pollution due to increased 180 atmospheric stability and transport of pollution from mainland Europe in the UK, respectively, and so care must be taken when comparing pre and post-lockdown levels of air pollution, as described in section 2.3.

### 3.2 Observed changes in daily mean and diurnal variations of NO$_2$

Measurements in 2020 from 65 urban traffic (figure S2a) and 61 urban background (figure S2b) AURN 185 measurement sites across the UK show clear reductions in NO$_2$ concentrations across all sites since the lockdown. Some of these differences are due to the natural seasonal variation in NO$_2$. To account for these expected variations, we calculate the daily difference of NO$_2$ values from 2020 with mean NO$_2$ values from detrended values from 2015 to 2019 for the appropriate day of year. This approach allows us to emphasize the difference of NO$_2$ values in 2020 from previous years. During the lockdown period, we find that 83% of days in 2020 at urban 190 traffic sites have lower NO2 values, far outnumbering those with higher NO$_2$ values (17%). During the pre-lockdown period, we find 76% of days at urban traffic sites in 2020 are below the 2015-2019 mean. We find a similar situation for urban background sites, with 73% of days above the 2015-2019 mean, but the decrease during the lockdown period is not as dramatic.

Figure 4 shows the percentage difference for all urban traffic and urban background sites for the lockdown period in 2020 compared to the same period averaged across 2015-2019. After removing site-dependent trends, it is observed that urban traffic sites have a mean decrease of 13.4 μg m$^{-3}$ in NO$_2$ over the lockdown period compared with the same period over the previous five years. This mean decrease approximately equates to a 48% drop in NO$_2$ levels across the UK. The AURN site Glasgow Kerbside observed the largest mean percentage decrease of 200 71.2% during the lockdown period, closely followed by Cambridge Roadside (68.8%) and Marylebone Road



(London) with a decrease of 67.8%. In total, 32 of the 65 urban traffic sites saw a decrease in $NO_2$ of greater than 50%. Armagh (Northern Ireland) is the only urban traffic site to show a mean increases in $NO_2$ 1.3 µg m$^{-3}$ (6.7%). Urban background sites show a smaller mean reduction of 4.9 (5.5) µg m$^{-3}$ , equating to a decrease of 40.6% in $NO_2$ levels across the UK. The largest decrease of 25.7 µg m$^{-3}$ was observed in London Hillingdon, corresponding

to 59.3%. Small increases (< 3 µg m$^{-3}$) were seen in York Bootham (Yorkshire and Humberside), and Eastbourne (South East). On average across all urban sites (traffic and background), a decrease in $NO_2$ of 42% is observed.

We perform independent z-tests on $NO_2$ measurements during the lockdown period and the mean from the same period from the previous five years, for $NO_2$ measurements during the lockdown period and measurements in

2020 immediately prior to the lockdown, and for $NO_2$ measurements immediately prior to lockdown in 2020 with the same period for the previous five years. We find that using these t-tests that 109 out of the 128 (85%) urban sites show a statistically significant ($p < 0.01$) difference in $NO_2$ between the mean observations during lockdown to the mean of the same period during the past five years. We also find 104 sites (from a possible 128 sites, 81%) show a significant difference between $NO_2$ measurements made in 2020 immediately prior to the lockdown to the

mean of the same period from the previous five years. The mean $NO_2$ difference for all urban background (traffic) sites for the time period immediately prior to the lockdown is -6.3 (-9.2) µg m$^{-3}$. We attribute this difference to changes in meteorology during January and February 2020 (Figure 3) when winds were higher than the previous five years. Finally, we find that 97 sites (75%) show a statistically significant difference between mean $NO_2$ observations during lockdown and immediately prior to lockdown in 2020, implying there was a significant drop

in $NO_2$ across the UK as a direct result of the lockdown.

We also examine mean changes in the diurnal cycles of $NO_2$ at urban traffic and urban background sites in London, Bristol, Cardiff, Newcastle, Glasgow and Belfast during the lockdown period compared to the same periods from the 2015-2019 mean. Based on the diurnal profile of $NO_2$ levels, we find that (Figure S3) typical pre-lockdown

diurnal cycles are driven by emission peaks in the morning and evening rush hours, with the evening peaks supressed due to the higher mean boundary layer that is grown during the day. In general, we find that the evening rush hour peaks at urban traffic sites across all cities during the lockdown period are suppressed compared to previous years, potentially due to changing working patterns. A notable exception is in Cardiff where the morning rush hour peak is suppressed. In contrast, at urban background sites in Cardiff the diurnal cycle of $NO_2$ is very

similar in 2020 to the previous years, with rush hour peaks of similar magnitude in the morning and evening.

### 3.3 Observed changes in daily mean $O_3$ and $O_x$ (= $O_3$ + $NO_2$)

Typically, close to sources of $NO_x$, $O_3$ is suppressed due to the reaction of high levels of NO with $O_3$. Further away from the sources, $O_3$ can reform through the oxidation of NO to $NO_2$ with peroxy radicals (formed from the

reaction of VOCs with OH) and subsequent photolysis of $NO_2$ to form $O_3$. To account for this photochemistry, we also report changes in the total oxidant, $O_X$, the sum of $O_3$ and $NO_2$, which should be approximately conserved in the absence of any change in the source strength of $NO_x$ or VOCs or a change in OH.

Figure S4 shows measurements of $O_3$ in 2020 from 46 urban traffic and background AURN measurement sites

across the UK, along with the daily difference of $NO_2$ values from 2020 with mean O3 values from detrended



values from 2015 to 2019. It shows the opposite trend to $NO_2$, with clear increases across the majority of the sites. Figure 4 shows the percentage difference for all urban traffic and urban background sites for the lockdown period in 2020 compared to the same period averaged across 2015-2019. After we remove site-specific trends, as described in section 2.3, we find that $O_3$ at urban background sites increased by a mean value of 7.15µg m$^{-3}$ during

the lockdown period when compared with the previous five years equating to a percentage increase of 11%. Leamington Spa (West Midlands) and London Hillingdon observed the largest mean increases of 21.3 µg m$^{-3}$ (35%) and 21.6 µg m$^{-3}$ (54%) respectively. Three sites observed a decrease during the lockdown with Aberdeen seeing a large decrease in $O_3$ of 24.0 µg m$^{-3}$ (36%) even though this site also experienced a substantial decrease in $NO_2$. We do not have a definitive explanation for this result, but it is consistent with a $NO_x$-limited

photochemical environment in which a decrease in $NO_2$ would reduce $O_3$ production. This could be achieved by possible fugitive emissions from the onshore gas terminals near Aberdeen. Only three urban traffic sites measured $O_3$ during our study period. Of those Marylebone Road (London) saw the largest increase (32.0 µg m$^{-3}$ or 104%) followed by Exeter Roadside (South West) with an increase 20.0 µg m$^{-3}$ (47%) and Birmingham A4540 Roadside (West Midlands) with an increase of 13.3 µg m$^{-3}$ (25%).


A similar statistical analysis has also been carried out for $O_x$ ($NO_2 + O_3$) and we find that a mean (median) increase of $O_x$ at urban background sites of 3.5 µg m$^{-3}$ (1.4 µg m$^{-3}$) or 3%. The two outliers are Leamington Spa (West Midlands) where we find the largest $O_x$ increase of 32.5 µg m$^{-3}$ and Aberdeen where we find the largest $O_X$ decrease of -27.6 µg m$^{-3}$. The three urban traffic sites measuring both $O_3$ and $NO_2$ show a large range in observed

differences of 4.2 µg m$^{-3}$ (+3%) at Birmingham A4540 Roadside, -7.9 µg m$^{-3}$ (-11%) at Exeter Roadside and -20.5 µg m$^{-3}$ (-15%) at London Marylebone Road.

Following our approach for $NO_2$, we use independent z-tests to determine the significance of changes in $O_3$ pre-lockdown and lockdown periods in 2020 and in the previous five years. We find that 36 out of 46 urban sites

(78%) show a statistically significant ($p < .01$) difference between the mean $O_3$ observations during lockdown to the mean of the same period from 2015 to 2019. However, we also find that 38 of those sites (83%) show a statistically significant difference between $O_3$ measurements immediately prior to the lockdown compared to the mean of the same period from 2015 to 2019. Finally, we find that 41 sites (95%) show a statistically significant difference between $O_3$ observations during the lockdown period and values taken from the period immediately

prior to the lockdown.

### 3.4 Relationship to emissions

During the lockdown period there has been around a 75% reduction in road traffic across the UK (see Figure 1). According to the NAEI, road transport is estimated to make up 53% of $NO_x$ emissions in the 1km x 1km grid

square that both urban background and urban traffic sites are situated in (Defra, 2018b). Therefore, we might expect there to be a reduction in $NO_2$ of around 40% across all sites. Mean decreases in $NO_x$ are very similar to those for $NO_2$ described above (47% at urban traffic, 40% at urban background – see figure S5), which is in line with the 40% reduction figure. However, it is clear however that individual sites have very different behaviour and the 75% traffic reduction may not necessarily equate to 75% reduction in emissions because different types

of vehicle were affected differently, with the most reduction in passenger cars and less reduction in high emitters



like HGVs. There is a wide range of contributions of $NO_x$ emissions from road traffic across the sites and there does not appear to be much correlation between this and the reduction seen during lockdown (see Figure S6), suggesting that the change in traffic flow near to individual sites is variable and will be to largest contributing factor to $NO_2$ reductions.


## 4 Discussion

### 4.1 Surface $O_3$

The COVID-19 lockdown has resulted in a significant decrease in $NO_2$ in cities across the UK, largely caused by the reduction of $NO_x$ emissions due to reduced traffic, and a concurrent increase in $O_3$. $NO_x$ and $O_3$ are closely

linked through their photochemistry and here we examine the reasons for the $O_3$ increase (Lelieveld and Dentener, 2000). A key factor that plays a role in $O_3$ concentrations is meteorology (Monks, 2000). High levels of actinic radiation cause the photochemistry involved in $O_3$ formation to happen faster and low wind speed conditions allow precursor species such as NOx and VOCs to build up and react to form $O_3$. Therefore, observed variations of $O_3$ in different UK cities will be influenced by a number of processes to varying degrees. Figure 6 examines $NO_2$, $O_3$

and total $O_x$ during the lockdown period for urban background sites in six different cities across the UK. Any change in $O_x$ can be thought of as a change in the abundance of oxidants, taking into account the repartitioning of $NO_2$ and $O_3$ caused by changes in $NO_x$ emissions. Whilst all cities have seen an increase in $O_3$ in the urban background compared to previous years, only southern UK cities saw a significant increase in total $O_x$, with London, Bristol and Cardiff showing a 5% increase. In contrast, $O_x$ slightly decreased in Newcastle, Glasgow and

Belfast. To assess OH being the cause of changes in $O_x$, we examine six measurements of total UVA at eight sites in the UK and compared data from 2020 to the mean of the previous five years (Figure S7). We find levels of UV across the UK were higher in 2020 compared to previous years, with the largest increases in southern UK. London, Chilton and Camborne saw increases of around 50% compared to previous years, with Glasgow and Inverness showing smaller increases of around 30%. Figure 7a shows a summary of the $O_x$ change in 2020 compared to

2015-2019 from individual sites across the UK as a function of latitude. We find a positive trend in $O_x$ towards lower latitudes, consistent with the higher excess UV levels further South. Figure 7b shows the temperature in 2020 compared to the previous five years as a function of latitude. Whilst these data are more scattered than for $O_x$, the period immediately prior to the lockdown was warmer than climatological mean values across the UK than the previous five years, with the largest increases in temperature at the lowest latitudes. In London, Bristol

and Cardiff, the increased temperature is around 2°C compared to previous years. Belfast and Glasgow did not see a significant temperature difference in 2020.

Observed variations in $O_3$ will also reflect changes in precursor VOCs. Online measurements of VOCs are only available at a small number of sites and here we consider measurements made at London Marylebone Road (an

urban traffic site) and London Eltham (an urban background site). Figure S8 shows measurements of a range of different VOCs for each site during 2020 and mean values for 2017-2019 when data are available at both sites. The data show most VOCs decrease in concentration during the post-lockdown period in 2020 compared to previous years. This is particularly true at the urban traffic site and for species such as benzene and toluene that have a largely traffic source and saw a decrease of 23% and 29%, respectively, compared to previous years. At

Eltham the decreases were both around 12%. 1,3 butadiene and 1-butene stand out as species showing a larger





increase in 2020 compared to previous years and the reasons for this are not immediately clear. VOCs have a wide range of lifetimes and emissions sources and they can be transported large distances, meaning their concentrations at a given site is much more affected by meteorology and chemistry than $NO_2$. When examining $O_3$ it can be useful to look at the total VOC loading and OH reactivity (k'). Figure S9 shows total VOC loading in ppb and total OH reactivity for each day in 2020, with colours showing the percentage change from the previous three years for that day. A full analysis of the behaviour of different VOCs during the lockdown period is beyond the scope of this work. Our focused analysis shows that while the picture is not straightforward, there is an apparent decrease in total VOCs at both sites compared to previous years. Mean values for total VOCs at Marylebone Road were 17% lower and the corresponding k' 15% lower than the 2017-2019 mean. At Eltham total VOCs saw a decrease of 10%, with a slight increase in the total k', largely driven by an increase in biogenically emitted isoprene.

Figure 8 shows daytime mean (10:00 – 18:00) isoprene data and its contribution to k' at two sites. Observed isoprene was a factor of two higher at both Marylebone Road and London Eltham during April and May 2020 compared to those months in previous years. Isoprene represents only a small contributor to OH reactivity at Marylebone Road, but at Eltham in 2020 it represents around 25% of total k'. Biogenic emissions of isoprene, originating from a variety of trees and shrubs, are driven in part by temperature and so it is perhaps not surprising that isoprene levels at the London sites were higher in 2020 compared to previous years due to the temperature increases described above. Further detailed chemical modelling studies, beyond the scope of this study, are required to assess in detail the chemistry behind $O_3$ formation and how this has been affected by the lockdown, however it is clear that $O_3$ has increased across the UK due to the reduction in $NO_x$, with an increase in total $O_x$ at Urban Background sites in the South of England. This is likely due to increased radiation and biogenically emitted VOCs compared to previous years, things that are unlikely to be linked to the COVID-19 lockdown.

## 4.2 Exceedances.

To put the changes in air pollutants in context with human health effects we have examined the number of exceedances of UK air quality objectives (Defra, 2019) and EU directive limits (EEA, 2016) for both $NO_2$ and $O_3$ in 2020 compared to previous years (see Table 1). For this analysis, we have used data from the London Air Quality Network (LAQN) consisting of 9 kerbside, 52 roadside, and 25 background sites for $NO_2$ and 1 kerbside, 8 roadside and 15 urban background sites for $O_3$ in the Greater London area. London has historically had by far the largest number of air quality exceedances in the UK so this analysis allows us to see the effect of the lockdown on the city with the most acute air pollution problems.

Exceedances were calculated on a per site basis, and then summed across all sites of a given type. For $NO_2$ a simple one-hour mean was calculated and each hour greater than 200 ug m$^{-3}$ was counted as an exceedance. We calculated a rolling mean value for $O_3$, using a window of eight hours and a step size of one hour. If a given calendar day saw this rolling mean exceed 100 ug m$^{-3}$, an exceedance was counted. Using this method multiple exceedances (contiguous or separated in time) were only counted as one to avoid ambiguity in their definition, and therefore can be thought of as "days when an $O_3$ exceedance occurred".



Figure 9 shows the results for January – May 2020 and comparisons to those months in 2015—2019. We find a general downward trend of $NO_2$ exceedances at roadside and kerbside sites in London, due to the continued reduction in $NO_x$ emissions from the vehicle fleet. At kerbside, the number of exceedances dropped quickly from 2395 in 2015 to only 28 in 2019. At roadside sites, exceedances dropped consistently from 472 in 2015 to 45 in

2019, with almost all of the remaining 45 at two sites in Wandsworth (Putney High Street) and Strand in Westminster. In 2020, up until the end of May, there was only one $NO_2$ exceedance at sites across the LAQN network, again at Putney High Street. Because we have only analysed data up until the end of May 2020, we do not know the cumulative effect on exceedances for the year or how many exceedances breach the 18 allowed by the air quality objective. Consequently, further analysis on data collected for the whole of 2020, including the

period when lockdown restrictions were relaxed, is required to put 2020 into context of previous year. As an estimate of the effect the lockdown may have on total exceedances in 2020, we replaced the number of exceedances during the lockdown period in 2019 with the number from 2020. This resulted in a 47 % decrease (34 to 18) in total exceedances of $NO_2$ at kerbside sites and a 12 % (76 to 67) at roadside sites. As the effects of lockdown certainly extend beyond the end of the time period explored by this study, we would expect there to be

less exceedances still during the remainder of 2020.

When considering any health benefits to this apparent improvement in air quality due to reduced $NO_2$ we should also consider exceedances to $O_3$ limits. The WHO has set a guideline value for ozone levels at 100 µg m$^{-3}$ for an 8-hour daily mean. Figure 9 also shows the total number of exceedances of this limit across kerbside, roadside

and urban background sites in the LAQN network for March – May in all years from 2015 - 2020. Urban background sites have seen a consistent increase from 5 in 2015 to 28 in 2018, followed by a drop to 18 in 2019 and an increase to 36 exceedances up until the end of May 2020. Peak $O_3$ in the UK often occurs in June and July so it will be necessary to analyse data from the whole year, alongside $NO_2$, in order to fully assess the importance but it is clear that any perceived benefits of reduced $NO_2$, both during the lockdown and in a lower $NO_x$ future,

should be considered alongside any concurrent increase in $O_3$.

### 4.3 Global comparison

Lockdowns to prevent the spread of COVID-19 have occurred globally and the effects of the change of emissions

on $NO_2$ and $O_3$ have been observed using satellite and in-situ measurements in several studies. Table 2 summarizes studies from a growing body of work that report changes in $NO_2$ and other air pollutants in countries across the world that are associated with the global COVID-19 lockdown, including satellite observations (Liu et al., 2020) and in situ measurements. These studies have used various methods to isolate the impact of the COVID-19 lockdown on changes in air pollutants from confounding factors, e.g. meteorology, using atmospheric chemistry

transport models and weather normalisation techniques based on machine learning (ML) algorithms. In Europe, reductions of $NO_2$ are typically slightly larger than we have seen in the UK in our study, perhaps reflecting more stringent lockdown policy. In Spain, $NO_2$ was reduced by 50% at both urban traffic and urban background sites (Petetin et al., 2020) and in Rome, Turin and Nice, $NO_2$ was reduced by 46, 30 and 63% respectively (Sicard et al., 2020). In all of these studies similar magnitude increases of $O_3$ were observed, mainly attributed to the

decreased NO. Further afield, in India TROPOMI satellite measurements showed that during the COVID-19





lockdown, there was a 18% decrease in $NO_2$ over the whole country, with a 54% decrease over New Delhi compared to the same period in 2015-2019 (Pathakoti et al., 2020). In situ measurements in New Delhi showed a 53% decrease in $NO_2$ and a 0.8% increase in $O_3$ for the lockdown period compared to the 2 weeks immediately preceding it. In Rio de Janeiro, Brazil, there was a 24-33% decrease in $NO_2$ during the lockdown compared to the

week before (Dantas et al., 2020) and in Sao Paulo data from urban roadside sites showed a 54% decrease in $NO_2$ compared to the previous 5 years (Nakada and Urban, 2020). In China, satellite observations showed a mean $NO_2$ decrease of 21% decrease across the whole country, relative to a similar period in 2015-2019 (Bao and Zhang, 2020). In situ measurements in cities in northern China measurements before and after lockdown showed a 53% decrease in $NO_2$ (Shi and Brasseur, 2020) and in-situ measurements in cities across the whole of China showed a

60% decrease in $NO_2$ comparing 1-24th January 2020 and 26th January - 17th February 2020 (Huang et al., 2020). Both these two studies also reported a >100% increase in $O_3$. These studies were both during wintertime so the $O_3$ increase was largely attributed to the reduction on NO emissions reducing titration of $O_3$ to $NO_2$, however the possible effect of reduced particles on UV radiation and hence $O_3$ production was also considered to have led to some of the increased $O_3$. Le et al, 2020 use satellite data to show a 71.9% decrease in $NO_2$ and 93% decrease in

Wuhan at the peak of the outbreak. They also report a 25.1% increase in O3 in Wuhan, largely attributed to a reduction in titration with NO. In the USA one study using EPA data showed a mean decrease of 30% of $NO_2$ in urban areas of Seattle, Los Angeles and New York during the lockdown. The study did not show any consistent change in $O_3$ levels (Bekbulat et al., 2020).

**5 Summary and conclusions**

We examined $NO_2$ and $O_3$ measurements from urban traffic and urban background sites across the UK during the COVID-19 lockdown period in 2020 ($23^{rd}$ March – $31^{st}$ May). We compared data to the detrended average from the previous five years in order to assess how these air pollutants have changed as a result of the reduced activity caused by the nationwide lockdown. $NO_2$ decreased by an average of 48% and 40% at urban traffic and urban

background sites, respectively. This is in broad agreement with the expected reduction based on the reduction in traffic and the proportion of $NO_x$ in the UK that comes from vehicles. For $O_3$, we find that values increased on average by 11% at urban background sites and by 48% at the three urban traffic sites. Total $O_x$ increased by 3% on average, suggesting the majority of the increase in $O_3$ is due to photochemical repartitioning as $NO_x$ is decreased. However there are difference across the UK, with the southern cities London, Bristol and Cardiff

showing a 5% increase in $O_x$ and Newcastle, Belfast and Glasgow showing only a slight decrease in $O_x$. Whilst anthropogenic VOCs are slightly decreased during the lockdown, we find some evidence that suggests that biogenic VOCs such as isoprene are higher due to warmer temperatures and higher UV levels across southern UK in 2020 compared to previous years; we find no evidence to suggest that higher UV levels were due to cleaner skies related to air pollution changes due to the lockdown. Analysis of exceedances of air quality objectives in

London for $NO_2$ and $O_3$ show that whilst there has been a decrease in exceedances of the $NO_2$ objective, this has come alongside an increase in $O_3$ exceedances. If we are to take the COVID-19 lockdown as an analogue of how air quality will respond to future reductions in emissions from vehicles (e.g. over the next 10-20 years), then this serves as a warning that $O_3$ must also be considered. In China, $NO_x$ reductions have led to increases in $O_3$ (Li et al., 2019a; Ma et al., 2016; Sun et al., 2016) and therefore air quality abatement strategies are being developed in

order to offset this, largely by also controlling VOCs (Li et al., 2019b; Le et al., 2020). In addition, a warming



climate is likely to cause increased emissions of biogenic VOCs, adding to the $O_3$ burden. Therefore it will be vital to control anthropogenic VOCs to avoid any health gains made by the reduction of $NO_2$ being offset by $O_3$ increases.

**Data availability**

The AURN data is all available for public download from the UK-AIR website (uk-air.defra.gov.uk). The LAQN data is available from the LondonAir website (londonair.org.uk). UVA data is available on request from Public Health England.

**Author contribution**

WSD and DPF carried out the data analysis and designed the figures. JDL and PIP wrote the manuscript with input from WSD and DPF. SEW designed and created figures and reviewed the manuscript.

**Competing interests**

The authors declare no conflict of interest.

**Acknowledgements**

The authors acknowledge Defra and uk-air.defra.gov.uk for provision of the $NO_2$, $O_3$, VOC and meteorological data, Public Health England for the UVA data and Ricardo PLC for temperature data. $NO_2$ and $O_3$ data for the

exceedance analysis is courtesy of the London Air Quality Network (operated by Imperial College London). JDL acknowledges the National Centre for Atmospheric Science for funding. WSD and SEW acknowledge the National Centre for Atmospheric Science for PhD studentships. JDL and WSD acknowledge the UK Natural Environment Research Council (grant # NE/T001917/1). PIP and DPF acknowledge the UK Natural Environment Research Council (grant # NE/R011532/1).

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



**Figures**


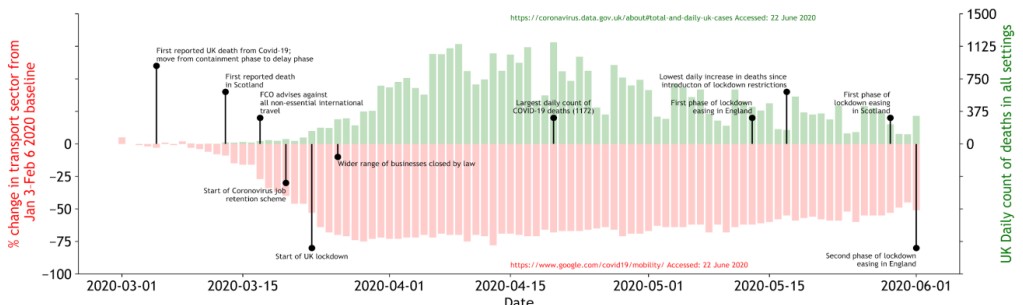

**Figure 1: Schematic of the timelines involved with daily mortality values attributed by the UK government to COVID-19, and changes in mobility from the transport sector (inferred from Google location data on smartphones) compared to a reference period (3 Jan- 6 Feb, 2020) before the lockdown period. Also included are key dates that describe the**

**run up and evolution of the UK lockdown. Data acknowledgements are shown inset of the plot.**






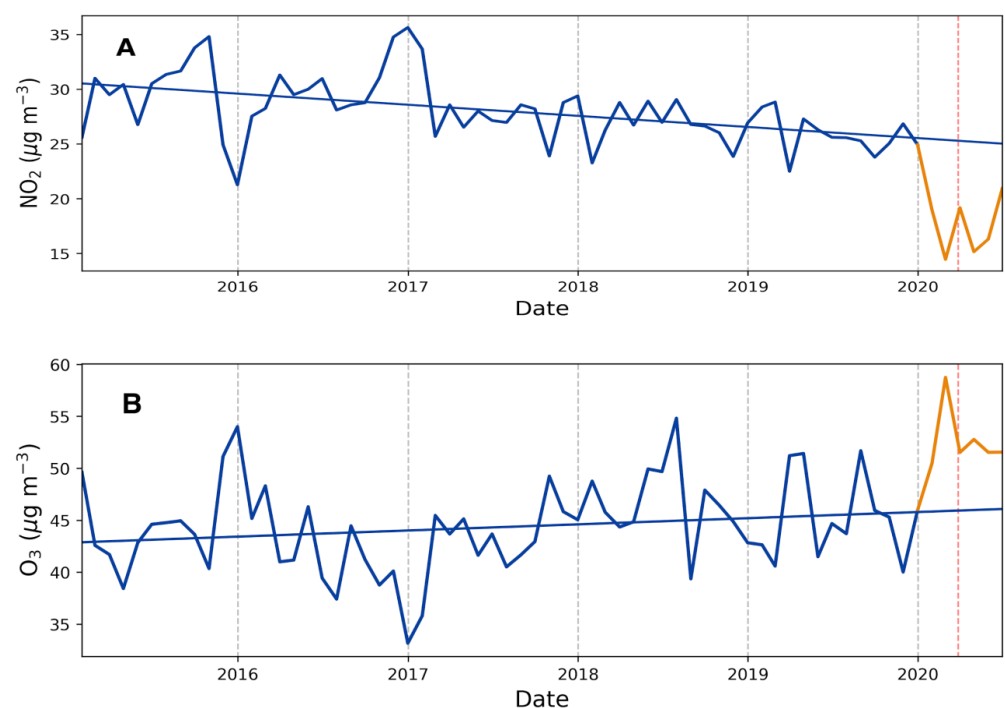

**Figure 2: Deseasonalized UK mean values of a) NO₂ and b) O₃ for all urban background and traffic sites from 2015 to 2020, with the mean 2015 -2019 trend superimposed. Data from 2020 is shown in orange, with the red dashed line**

**denoting the start of the lockdown on 23th March 2020.**



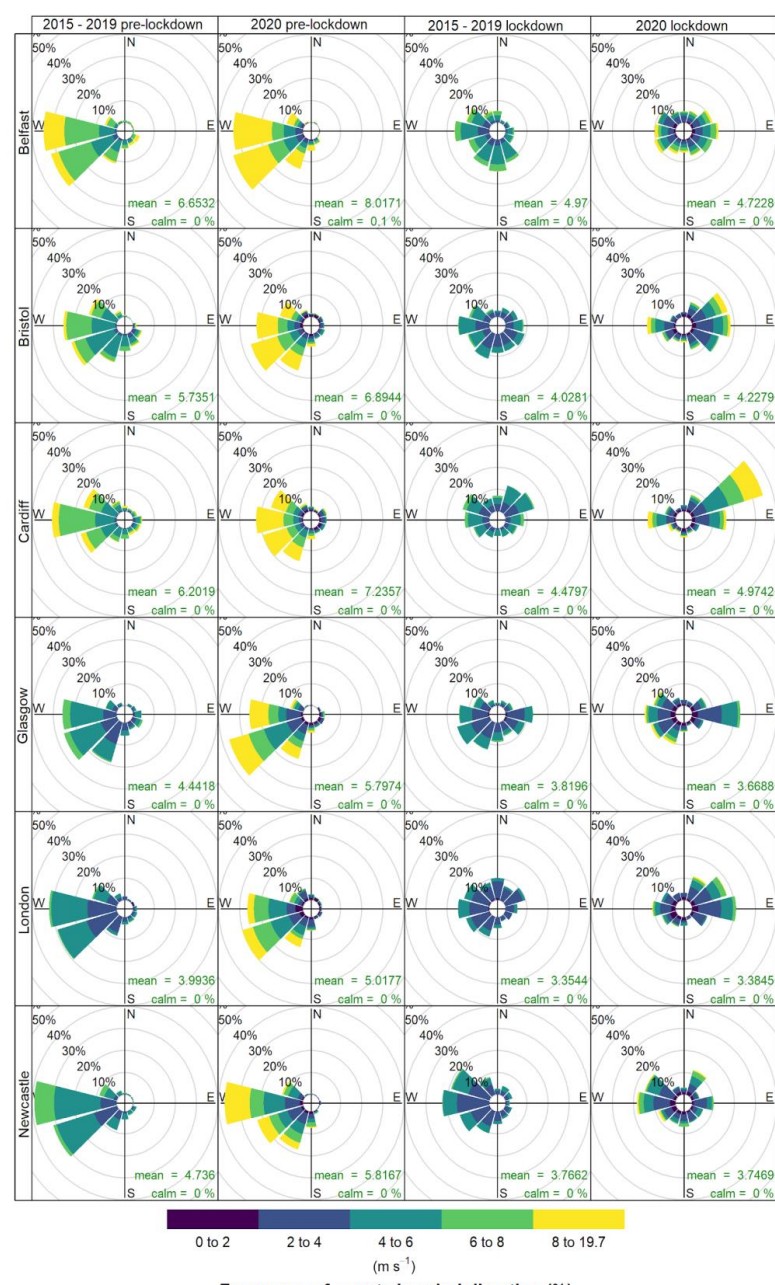

**Figure 3: Average wind roses for 6 cities for pre and post lockdown period and lockdown period 2015-2019 and 2020.**

**Data used is modelled using the UK Met Office Unified Model.**



**Figure 4: Percentage change in NO₂ at all urban background and urban traffic sites for the lockdown period (23rd March – 31st May) in 2020 compared to the same period averaged across the previous 5 years, after removing site-dependent trends. The lighter coloured bar at the top shows the average of all sites. Site acronyms can be found in the SI.**





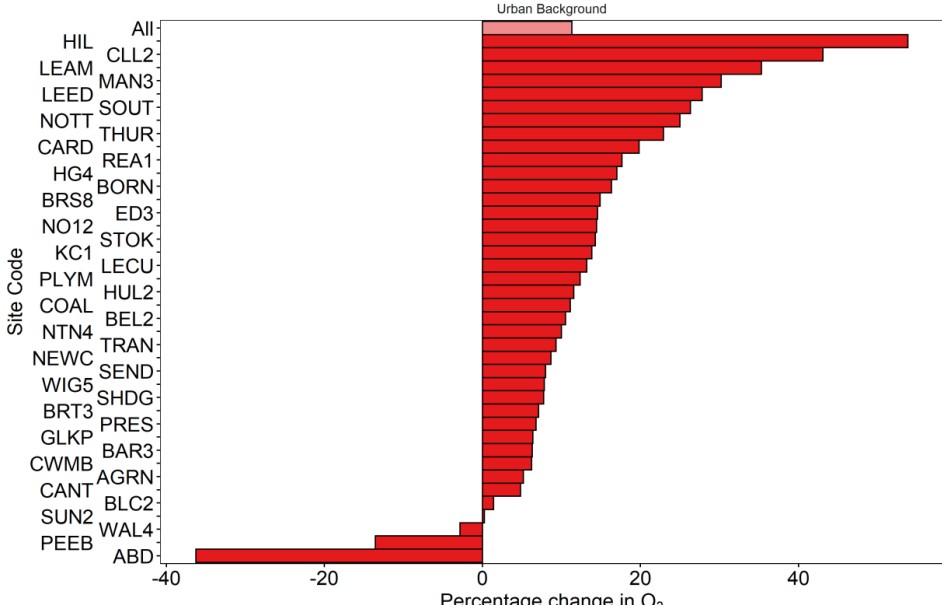

**Figure 5: Percentage change in O₃ at all urban background sites for the lockdown period (23ʳᵈ March – 31ˢᵗ May) in 2020 compared to the same period averaged across the previous 5 years, after removing site-dependent trends. The lighter coloured bar at the top shows the average of all sites. Site acronyms can be found in the SI.**




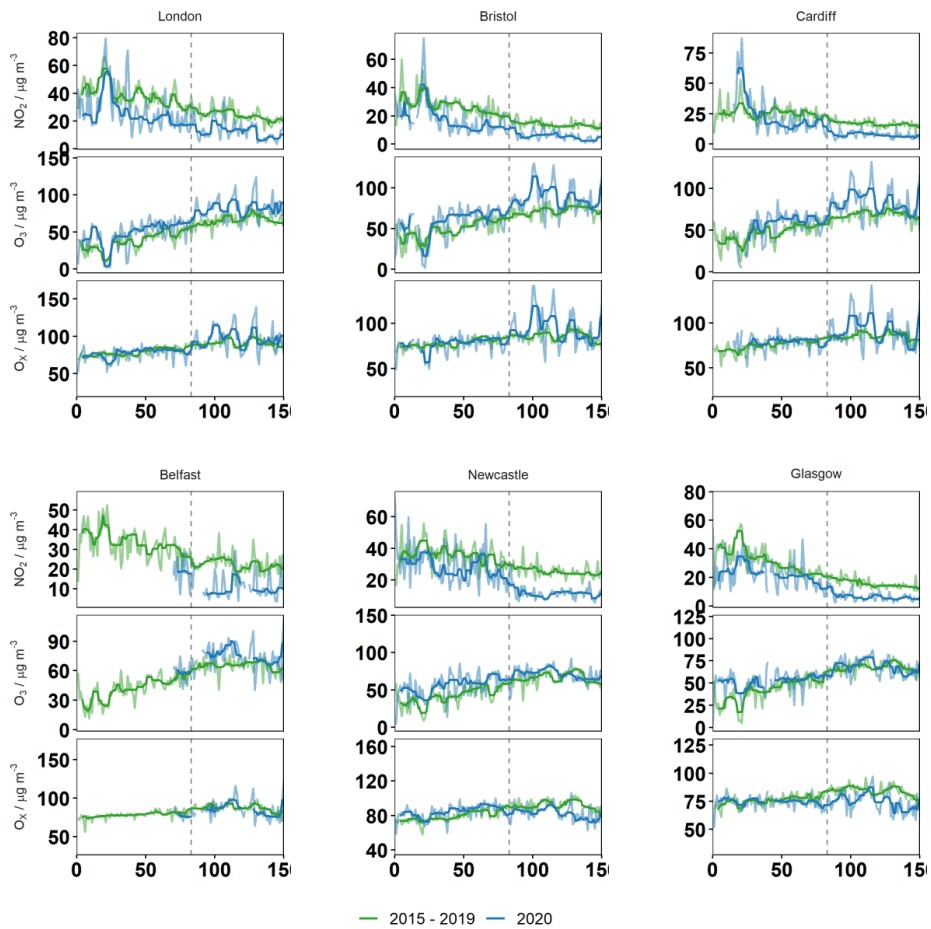

**Figure 6: Daily median time series of NO₂, O₃ and Oₓ (NO₂ + O₃) for 2020 and the average of 2015-2019 at urban background sites in 6 cities representing a geographical and political spread across the UK. The thick line represent 7 day rolling mean. The hashed grey line indicates the start of the lockdown period.**







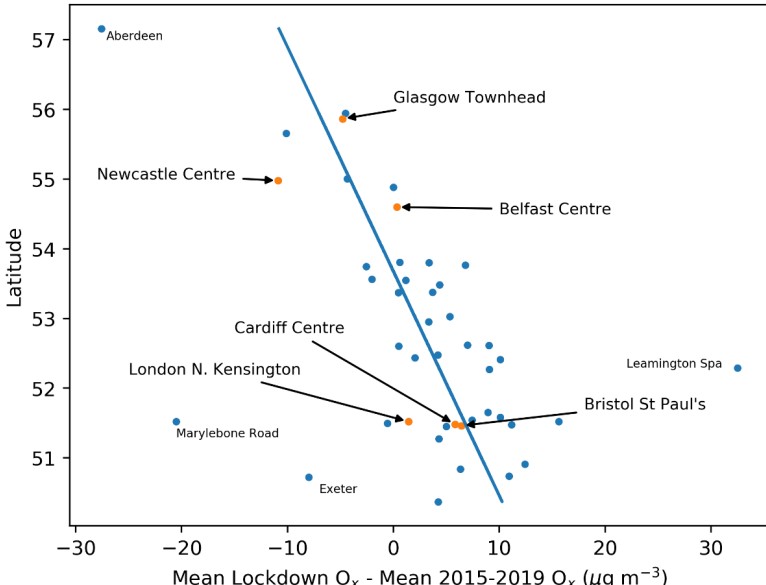

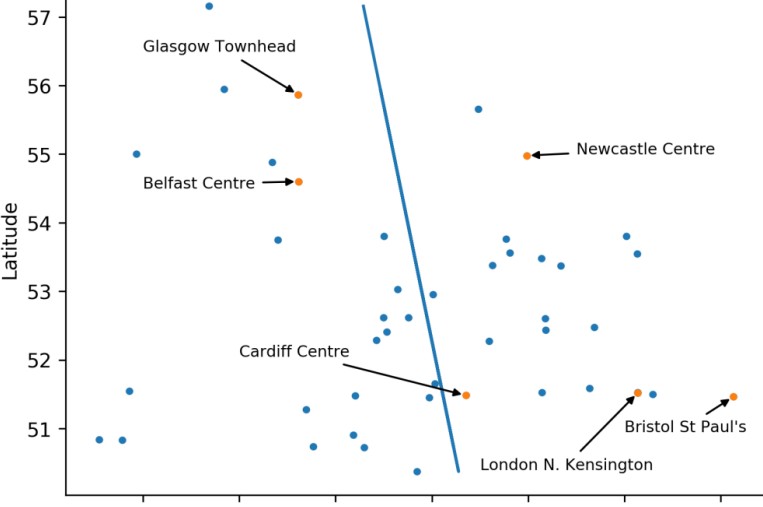

**Figure 7: Difference in mean (a) $O_x$ (µg m⁻³) and (b) Temperature between the lockdown period and the detrended mean of the same period from 2015 to 2019 for urban background sites as a function of latitude. Sites examined in figure 6 are highlighted in orange.**




**a.)**

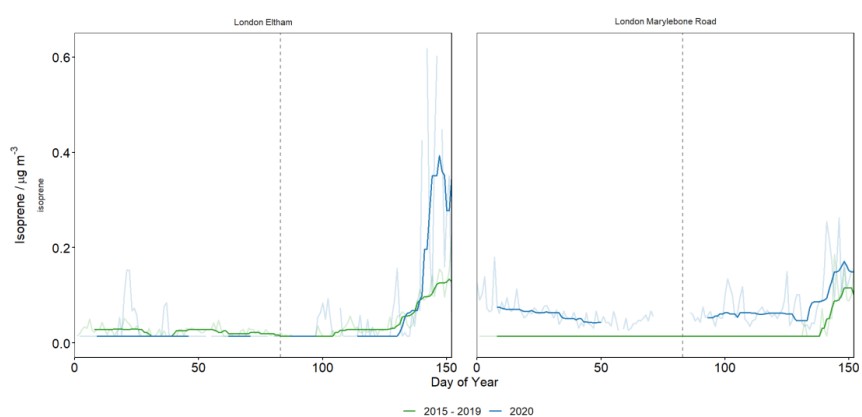

**b.)**

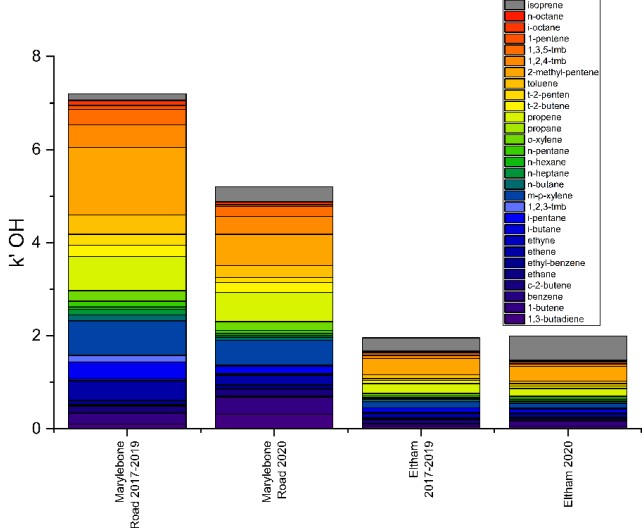

**Figure 8: (a) Levels of isoprene at the London Eltham (urban background) and London Marylebone Road (urban traffic) sites in 2020 and the average of 2017-2019. (b) shows the contribution of isoprene (grey slice) and other VOCS to total OH reactivity (k') for each site for 2017-19 compared to 2020.**




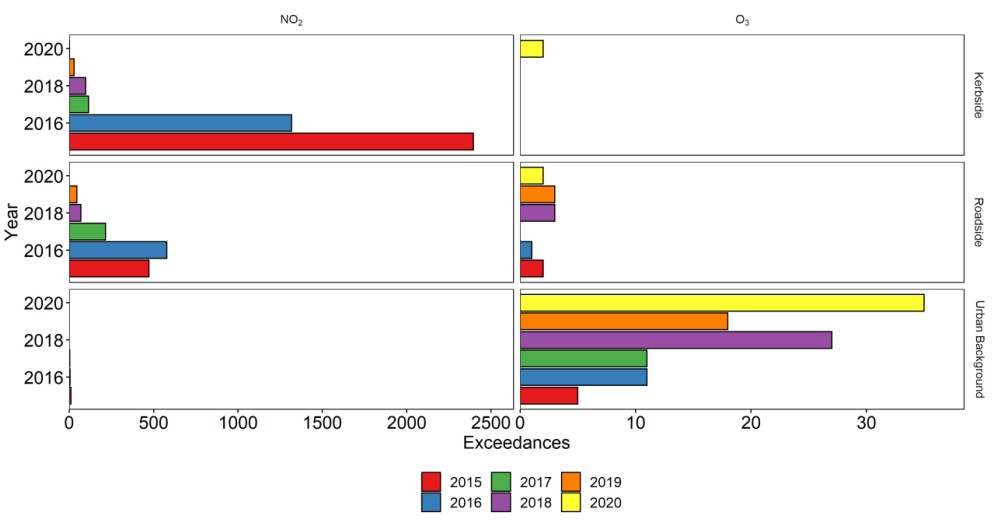

Figure 9: Exceedances of the UK air quality objectives for NO₂ and O₃ across the London Air Quality Network.





**Tables**

**Table 1. UK Air quality objectives (Defra, 2019). Note that the UK has adopted the EU NO$_2$ limit as a part of its air quality objectives, but improves upon the O$_3$ obligations where O$_3$ must not exceed 120 ug m$^{-3}$ more than 25 times per year in a given 3 year window (EEA, 2016).**

| Pollutant | Limit / ug m$^{-3}$ | Measured as | Allowed annual exceedances |
|-----------|---------------------|-------------|----------------------------|
| NO$_2$ | 200 | 1 hour mean | 18 |
| O$_3$ | 100 | 8 hour mean | 10 |





**Table 2: Summary of previous measurements.**

| Focus region | Observed change in $NO_2$ (NO) | Observed change in $O_3$ | Comments | Reference |
|---|---|---|---|---|
| **UK** | | | | |
| *UK wide* | 48% at Urban Traffic, 41% at Urban Background | 11% increase at urban background sites | Changes relative to detrended lockdown period 2015-2019. | *This study* |
| *UK wide* | -30 to -50% in urban areas | Increase mostly explained by reduced NO | UK government (Defra) synthesis report describing contributions from 50 individual responses. Data submitted up to 30[th] April 2020 | https://uk-air.defra.gov.uk/library/reports.php?report_id=1005 |
| **Europe** | | | | |
| *Greece* | -22% for March and April 2020 compared to 2019 | | TROPOMI monthly mean tropospheric nitrogen $NO_2$ observations used. | Koukouli et al. (2020) |
| *France* | -63% (-71%) Nice | +24% Nice | | Sicard et al. (2020) |
| *Italy* | -46% (-69%) Rome, -30% (-53%) Turin | +14% Rome, +27% Turin | | Sicard et al. (2020) |
| *Spain* | Mean changes over all three phases of the lockdown: -4.1 ppb (-50%) for background urban and –6.3 (-50%) for traffic sites. | -- | Used a ML approach to determine deviation from BAU NO2, trained using 2017-2019 data from background and traffic surface AQ monitoring sites. Study considers all three phases of lockdown up to 24[th] April 2020. | Petetin et al. (2020) |
| *Spain* | -69% Valencia | +2.4% Valencia | Hourly data provided by local and regional agencies. Changes relative to 2017-2019. All sites noted a decrease before the lockdown. Larger reductions observed at traffic sites. | Sicard et al. (2020) |
| **International** | | | | |
| *Brazil* | -24-33% compared to 2019 | -- | Study over Rio de Janeiro used data from automatic monitoring station run by Municipal Department of the Environment. Study period is from March 2[nd] to April 16[th] 2020, with lockdown on 23[rd] March 2020. | Dantas et al. (2020) |
| *Brazil* | -54% (-77%) on urban roads | +30% | Study over Sao Paulo using three in situ AQ sites. Changes relative to similar periods from previous five-year mean. | Nakada and Urban (2020) |
| *China* | -25% | -- | Study of data from 44 cities in northern China from 1[st] January to 21[st] March 2020. Lockdowns started on 23[rd] January in Wuhan with other cities following soon afterwards. Linear regression was used to determine BAU. | Bao and Zhang (2020) |
| *China* | -21% | | Study used satellite observations of tropospheric $NO_2$ data over China. | Liu et al. (2020) |



| | | | Decrease relative to similar period during 2015 to 2019. | |
|---|---|---|---|---|
| *China* | -53% | +100% | Study focused on northern China using in situ measurements. Data compared before and after lockdown. | Shi and Brasseur (2020) |
| *China* | -57% (-62%) | +36.4% | Study over Wuhan. | Sicard et al. (2020) |
| *China* | -60% | >+100% | Used in situ measurements across China. Differences between 1-24th January and 26th January-17th February 2020. | Huang et al. (2020) |
| *China* | -71.9% | +25.1% | TROMPI measurements over Eastern China and Wuhan, compared to previous 5 years. | Le et al. (2020) |
| *India* | -18% from previous 5-year mean; over New Delhi -54% | -- | Used satellite observations of NO2 from TROPOMI, relative to same period 2015-2019. | Pathakoti et al. (2020) |
| *India* | -53% over New Delhi compared to before lockdown. | +0.8% over New Delhi compared to before lockdown. | Using 34 monitoring in situ monitoring stations over New Delhi. Study compare pre-lockdown period 3rd-24th March and during lockdown period 24th March-14th April 2020. | Mahato et al. (2020) |
| *Kazakhstan* | -35% | +15% | Study over Akmaty using data from a similar previous period from 2018-2019. Data from the Airkaz publics AQ monitoring network. | Kerimray et al. (2020) |
| *Morocco* | -96% | -- | Study over Salé City, Morrocco using urban in situ data. | Otmani et al. (2020) |
| *USA* | -30% | Weak, inconsistent response | Used EPA data. | Bekbulat et al. (2020) |
