# Peer review of "UK surface NO2 levels dropped by 42% during the COVID-19 lockdown: impact on surface O3"

_Atmospheric Chemistry and Physics, 2020_

## Referee Comment (RC1) · Anonymous Referee #1 · 6 Sep 2020

General Comments:

Lee et al. analyze NOx and O3 data at a large set of routine air quality monitoring sites across the UK to assess changes in emissions and chemistry associated with the COVID-19 lockdowns. Their analysis runs through the spring of 2020, so not through the end of the lockdown period, but through the largest reductions in mobility. They compare NO2 and O3 during the lockdown to the historical average from the 5 previous years, and to a projection for the lockdown period based on a fit to the trend in the 5 year data set. They quantify changes in NO2, O3 and Ox based on this analysis, and show that large decreases in NO2 were offset by large increases in O3 at relatively constant Ox at sites across the UK. Further analysis of trends in UV radiation, temperature and isoprene (at only limited sites) showed that any apparent changes in total Ox were

likely related to these variables than to a response to emissions reductions.

Overall the paper will be of interest to ACP and should be published with revisions. The major comments that the authors should address are as follows.

1. The authors make reference to the influence of meteorology and invoke it to explain aspects of the data set qualitatively, but they refrain from a quantitative assessment of the role of meteorology in the main conclusions. The analysis requires either that they look at the relationships to meteorology in a more quantitative sense, or that they provide some quantitative set of uncertainties associated with neglecting the influence of meteorological variability. There are more comments to this effect below.

2. In several instances (e.g., abstract, $NO_2$ reduced by 42%), quantitative measures of the changes in air pollutants are quoted without error estimates or even measures of variability. Such estimates are required, especially if the work is to be compared to the large body of developing literature using different methodologies on this topic. It is unlikely that the numbers quoted here are exact. Again, there are more comments to this effect below.

3. In a number of instances, either the figures or the conclusions drawn from them are not clear. See comments below to improve readability and robustness of conclusions.

In addition to these comments, the authors should address the following more specific comments.

Specific Comments:

Line 41-42: What is the recent trend in $NO_x$ concentrations?

Line 48: The definition is for diameter rather than radius

Line 49-50: Check sentence grammar

Line 52-55: Statement may apply to urban centers, but it is not broadly true. $O_3$ has decreased with decreasing $NO_x$ emissions in many locations. See, for example:

Strode, S.A., J.M. Rodriguez, J.A. Logan, O.R. Cooper, J.C. Witte, L.N. Lamsal, M. Damon, B. Van Aartsen, S.D. Steenrod, and S.E. Strahan, Trends and variability in surface ozone over the United States. Journal of Geophysical Research: Atmospheres, 2015. 120(17): p. 9020-9042.

Cooper, O.R., D.D. Parrish, J. Ziemke, N.V. Balashov, M. Cupeiro, I.E. Galbally, S. Gilge, L. Horowitz, N.R. Jensen, J.-F. Lamarque, V. Naik, S. Oltmans, J. Schwab, D.T. Shindell, A.M. Thompson, V. Thouret, Y. Wang, and R.M. Zbinden, Global distribution and trends of tropospheric ozone: An observation-based review. Elem. Sci. Anth., 2014. 2: p. 29.

Cooper, O.R., R.-S. Gao, D. Tarasick, T. Leblanc, and C. Sweeney, Long-term ozone trends at rural ozone monitoring sites across the United States, 1990–2010. J. Geophys. Res., 2012. 117(D22): p. D22307.

Line 65, Figure 1: The notation in the insets is quite difficult to read and will not be legible on a printed page (readability required >200% magnification on my screen).

Line 66: Figure 1 uses Google mobility data, which are qualitative at best and have shown different reductions relative to other markers, such as traffic counts, in different regions. Can the authors provide another data set, such as traffic counts, or else some statement of uncertainty in the Google mobility data? If not, a caveat should appear re: the use of these data and their uncertainty as a proxy for actual traffic counts. Google and Apple mobility data are easy to obtain but not necessarily the best measure.

Line 115-116: It is useful to have the description of the NOx measurements in this paper and acknowledgement of the potential for interference on Mo converters – several recent analyses of COVID impacts have not addressed this issue at all. For this last statement, however, it is apparent that many of the sites in the network are not urban. What is the bias in using Mo converter NOx for these sites? Would this really be consistent with the 15% accuracy quoted below? How much does this affect the analysis of Ox later in the manuscript? As with other comments, this uncertainty should be

propagated through the quantitative measures given later.

Line 133-137: What variables were used for the detrending? Was this purely a seasonal trend, or were standard meteorological and day of week variables used? How well did this fit the 5 year data? The statement regarding emissions changes lacking importance during the 5 year period does not seem consistent with the two noted changes in 2017 and 2019. Would large step changes in emissions be expected, especially if new control measures were implemented in 2019, the year before the lockdowns?

Line 139, Figure 2: What is the measurement frequency or averaging period for the NO2 and O3 data? These appear to be approximately monthly averages? If data are heavily averaged, the figure should also show a variability? Such variability would represent both the variability in averaging to a monthly (?) value at a given site, and the averaging of all sites. It is also not clear that averaging this collection of sites together makes sense, since there are likely large, systematic difference between sites? A relative, rather than absolute, y axis would seem to be more appropriate in this case.

Line 180-181: "Care must be taken" in the comparison due to the meteorological variability. However, there does not appear to be an effort in this paper to correct the data for meteorological variability during the lockdown period. Perhaps this will appear in a later section, but some estimate, at least, of the effect of meteorological differences on the uncertainty of the NOx and O3 changes quoted in the paper is needed.

Line 184-185: Figure S2 is quite difficult to read, even with magnification. The color legends are not legible, and the red lines are nearly invisible. So despite the statement here that the figure shows clear changes, it is not of sufficient quality to do so.

Line 189-194: Were there any obvious differences in meteorology (e.g., just T for example) on the days that were above and below the long-term average?

Line 206 and Figure 4: What is the error bar on the 42%? See comments above – this should be shown in the figure and represent the site to site variability together with the

uncertainties in the measurements and potential meteorological artifacts.

Line 211: Is it a t-test or a z-test? Use consistent notation and give a definition. If the NO2 data are not normally distributed (which is likely), is the test appropriate?

Lines 214-220: The discussion here is somewhat unsatisfying, again due to the qualitative use of meteorology, which is rather arbitrarily applied to provide justification of a difference when one is not expected, but is not given any weight in the case where a difference is expected due to the lockdown.

Line 239, Figure S4: Same comment as above for Figure S2.

Line 242: Reference is likely intended to Figure 5 rather than 4.

Line 247-250: The previous discussion cites only the effect of titration by NO emission in regulating the response of O3. Here photochemistry is invoked. Is the seasonal photochemistry expected to be strong in this season in the UK? For example, does O3 exceed its background values at this time of year? The discussion re: petrochemical emissions seems quite speculative in the absence of measurement or modeling.

Line 273: Same comment as above re: the use of Google mobility data as a proxy for traffic. Caveats or uncertainties required, but more reliable data sets, such as actual traffic counts, would be preferable.

Line 288-311 and Figures 7 and S7: A 5% increase in Ox in southern UK cities is cited here. Similar to comments above, there is no statement of the uncertainty or variability in this estimate, but the authors should provide one. The changes in Ox appear to be well correlated with changes in UVA for 2020 – in other words, that the increased Ox is plausibly not attributable to reduced NOx emissions. This would also be consistent with the trend shown in Figure 7a, at least qualitatively. Is that the conclusion of this paragraph? It is not clear what is being said here. Finally, the T correlation in Figure 7b is difficult to interpret. What is the line? Is this a fit? If so, it does not appear to represent the data. The T data appear to be more related to the isoprene data in the

following section, and so perhaps should be presented with that figure.

Line 321-323: Statement might be true, but it would really depend on the dominant source for VOCs. For example, if VOC were mainly from traffic emission, as NOx emissions are stated to be for this region, then one might expect similar changes in the two. Absent some statement of an inventory and major VOC sources, this statement does not appear justified.

Line 333, Figure 8: Why is isoprene apparently exactly zero in the historical average at Marylebone Rd. for most of the record? Is this a measurement artifact?

Section 4.3: The literature review given here is useful, but it more likely introductory material than it is a conclusion. Suggest moving to the introduction.

Line 436-438: The "warning" regarding O3 is an important statement, but it is not clear that it is justified. The major influence of NOx, if I understand the conclusion of the paper correctly, is in the change in O3 titration, but not in photochemistry (at least not in the winter-spring season studied here). Thus, O3 would go to its background value in the absence of NO titration, while photochemistry would not be strongly affected. Does this scenario really constitute a "warning" that would need to be taken into account to inform emissions reduction policy? The finding is quite different from that referenced in the following sentence regarding O3 in China, where changes are clearly attributable to photochemical processes. The paper should not conflate the two regions, which are clearly quite different.
* * *

---

## Referee Comment (RC2) · Anonymous Referee #2 · 9 Sep 2020

The authors address the impacts of the COVID-19 lockdown to NO2 emission reductions in the UK, and the possible implications to surface O3 levels. More specifically, they present measurements from 128 urban monitoring stations and compare the 2020 lockdown period, to the 2020 pre-lockdown period, and the same periods from 2015-2019. They follow an approach to deseasonalise and linearly detrend the 2020 data based on the previous years to show that NO2 levels have dropped for various UK cities while O3 has increased.

Although the authors discuss the implications of meteorology to the NO2 concentration reductions these effects are not carefully taken into account. They present meteorological differences between these periods that show higher wind speeds during the lockdown and many times from different directions. A characteristic example is e.g.

Cardiff that showed increased wind speeds (Fig. 3) and the highest NO2 reductions (Fig. 4) that are currently fully attributed to the lockdown. I would recommend that the authors perform a more detailed analysis of the meteorological conditions and only include the cities that had similar wind speeds, wind directions, and exclude the ones that did not.

When moving to the O3 trends things become more challenging since O3 is strongly affected by meteorology as well as NOx and VOC emissions (the lifetimes of the latter also affected by meteorology). Although O3 formation is complicated the authors seem to oversimplify it and often promote a link between O3 formation and NOx reductions that is not supported at all by the observations. On the contrary, observations promote differences in the UV levels that could drastically increase O3 compared to previous years. Overall, the manuscript would be suited for ACP after (1) a careful exclusion of cities that had different meteorological conditions from 2020 to 2015-2019, and (2) more honest and precise conclusions regarding the increased O3 levels.

Specific comments

Page 1, lines 17-18: ". . . suggesting the majority of this change can be attributed to photochemical repartitioning due to the reduction in NOx.". The authors did not quantify the effect of meteorology and NOx reductions to be able to conclude this. Please rephrase.

Page 1, line 21-22: Can the authors make this statement without looking in more detail the meteorological differences between the studied years?

Page 1, line 37-39: Where is the remaining 16% NOx coming from? Please provide in parenthesis the variability as $\pm$ XX%. Also, there is no contribution of biomass burning to NOx which especially in the wintertime could play a role.

Section 2.3, line 133-134: ". . . we first linearly detrend and deseasonalise NO2 data at each AURN site based on the climatology of the previous five years". Please, elaborate

more and show characteristic examples of data before and after deseasonalising in the main text or SI. It was not clear to me what is shown in Fig. 2 and I had to spend a long time before understanding the de-seasonalisation approach (not 100% sure I still do). This is an important step for this study and is only very briefly discussed. This also includes the associated uncertainties.

Section 2.3, line 147: It is surprising to me that this sudden drop in January-February is suggested to be only due to emerging crises in nearby European cities. The authors later discuss that meteorology is significantly different for these months compared to March-May but still not that drastically different compared to the same months from previous years (Figure 3). I consider it important to understand where this drastic drop in concentration before the lockdown even started, is coming from. This rapid change not related to the pandemic is strong proof that this approach may not work since the needed weight to meteorology or other factors is not accounted for. If differences in meteorology between the 2015-2019 pre-lockdown, and the 2020 pre-lockdown are the reason for this drop in NO2 concentrations (which I suppose mostly is as also discussed in section 3.1) then similar differences during the lockdown (e.g. Cardiff) could play a crucial role in reduced NO2 concentrations.

Section 3.2, line 186-189: The authors already showed how strong influence meteorology could have on the trends based on the pre-lockdown period. If a comparison for the different years was made it should be followed (and weighted) by a comparison of wind direction, wind speeds. For example, Cardiff that has higher wind speeds in 2020 compared to other years (Figure 3) has the highest drop in NO2 which is not due to the lockdown alone. Also, it would be great to see the bars in Figure 4 colored based on the concentrations observed at each site, and with error bars.

Line 209: Is this the mean of all 4 years from 2015-2019? I wonder whether it would make more sense to compare only to 2019. More detailed sensitivity analysis and discussion will improve the presented results here and show whether uncertainties are higher than the observed trends.

Line 227-230: What is the contribution of biomass burning to NOx? The increase in the later hours promotes the possible effects of residential heating. Please discuss the contribution of other emission sources further in the main text.

Section 3.3, line 250: Photochemistry is a key driver for O3 production. However, the authors here don't address the possible effect of yearly variations in photochemistry. Comparing j-NO2 for the different years during these periods would be essential to answering this.

Line 288-311: Aren't the authors suggesting here that the increased O3 is mostly due to meteorology? Please emphasize this more and de-emphasize the O3 increase due to NOx reductions since there is no trend to support this.

Line 313-331: Various sources of VOCs and oxygenated VOCs are not discussed here, e.g. biomass burning, volatile chemical products, industry, that can play a crucial role in determining the total VOCs and total reactivity, and therefore understanding O3 formation. Presented here is not the total VOCs or total reactivity since the discussed VOCs are predominantly related to combustion/traffic emissions. In general, please emphasize more the variability of VOC emissions and that to understand O3 formation NOx and VOC emissions are equally important.

Line 341: Nothing is clear based on the presented results. The authors have no proof that O3 increased due to changes in NOx or changes in meteorology or VOCs. Please rephrase.

Figure 9 is since January although the lockdown was not in effect. How many exceedances happen during the pre-lockdown period? Please separate the two periods and further discuss them if necessary.

Line 427: The increase in Ox can be due to differences in UV levels that will increase OH and O3 levels as mentioned by the authors in the main text. Please rephrase.

Line 436-438: This is a stretch when there is no quantification of the factors affecting

[Figure]

O3 formation. Please rephrase.

Line 441-443: Strong wording. Please rephrase.

Technical comments:

Page 2, line 49: O3 is the main pollutant for urban pollution too. Please rephrase.

Page 2, line 78: Change "has" to "could have".

Page 3, line 96: correct to "levels are".

Page 4, line 121: delete "and". Also, an error is provided for the PM2.5 measurements but there is no mention of the type of instrumentation used. Since PM2.5 is not used at all in this study the authors could completely skip this.

Line 202: correct to "increase".

Line 213: Do you mean "Observed variations in O3 will also reflect changes in precursor VOC emissions"? Even then, how would that happen? Please rephrase.

Line 240: correct "O3".

Line 278: delete "however".

Figures comments:

Please improve the quality of the figures in the main text and supplement. Also, include uncertainties/error bars to the figures.

Figure 2: Could the authors add the 25th and 75th percentile? Also, could the authors present the results for urban and background environments in the SI for cases where this approach works and cases where this approach is more challenging?

Figure 6: x-axis label is missing.

---

## Author Response (AR1)

This document includes authors' responses to both reviewer 1 and reviewer 2. Reviewer's comments are in standard text. Authors' responses are bold text with any text taken from the manuscript in italics. New additions to the manuscript in response to reviewer comments are shown in red italics.

**Reviewer 1**

General Comments: Lee et al. analyze NOx and O3 data at a large set of routine air quality monitoring sites across the UK to assess changes in emissions and chemistry associated with the COVID-19 lockdowns. Their analysis runs through the spring of 2020, so not through the end of the lockdown period, but through the largest reductions in mobility. They compare NO2 and O3 during the lockdown to the historical average from the 5 previous years, and to a projection for the lockdown period based on a fit to the trend in the 5 year data set. They quantify changes in NO2, O3 and Ox based on this analysis, and show that large decreases in NO2 were offset by large increases in O3 at relatively constant Ox at sites across the UK. Further analysis of trends in UV radiation, temperature and isoprene (at only limited sites) showed that any apparent changes in total Ox were likely related to these variables than to a response to emissions reductions. Overall the paper will be of interest to ACP and should be published with revisions.

**We thank the reviewer for the detailed review that will no doubt improve the manuscript.**

The major comments that the authors should address are as follows.

1. The authors make reference to the influence of meteorology and invoke it to explain aspects of the data set qualitatively, but they refrain from a quantitative assessment of the role of meteorology in the main conclusions. The analysis requires either that they look at the relationships to meteorology in a more quantitative sense, or that they provide some quantitative set of uncertainties associated with neglecting the influence of meteorological variability. There are more comments to this effect below.

2. In several instances (e.g., abstract, NO2 reduced by 42%), quantitative measures of the changes in air pollutants are quoted without error estimates or even measures of variability. Such estimates are required, especially if the work is to be compared to the large body of developing literature using different methodologies on this topic. It is unlikely that the numbers quoted here are exact. Again, there are more comments to this effect below.

**We deal with both of these comments together. We chose not to do a quantitative assessment of the role of meteorology but to compare to data from the previous 5 years. As stated in the text** "*we acknowledge there are uncertainties associated with our approach, but this method offers simplicity and straightforward error propagation*". **However the reviewer is correct we neglected**

to include an error analysis for our approach and we have now added this.  We combined the standard error in the median of the daily median concentrations for the lockdown period in 2015-2019 and 2020. This is now shown as error bars on figures 4 and 5 and every time we quote a change in pollutant levels in the text we now quote an error. We hope this gives a better description of the changes observed, without the need for a full analysis of the meteorology (such as that done in Grange and Carslaw 2020), which would also be subject to errors.

We added the following sentence to the text (line 235):

*To assess the error, we combined the standard error in the median of the daily median concentrations for the lockdown period in 2015-2019 and 2020, with error bars shown on the graph.*

**Figures 4 and 5 are now:**

**Figure 4:**

[Figure]

**Figure 5:**

[Figure]

3. In a number of instances, either the figures or the conclusions drawn from them are not clear. See comments below to improve readability and robustness of conclusions.

**We deal with the specific comments below.**

In addition to these comments, the authors should address the following more specific comments.

Specific Comments:

Line 41-42: What is the recent trend in NOx concentrations?

**We show this for the different site types later (line 159-164) but we have averaged these a see a 3.3% per year decrease in NO$_x$. The sentence now reads:**

*In recent years, there has been a pronounced reduction in NOx emission (Defra, 2018a) that largely reflects lower transport emissions, with NO2 showing an average decrease of 3.3 % per year since 2015.*

Line 48: The definition is for diameter rather than radius

**We have changed this.**

Line 49-50: Check sentence grammar

**The sentence now reads:**

*At present the main pollutants of concern in urban centres are NO2 and particulate matter with radii diameter smaller than 2.5 microns (PM2.5) in urban centres, and in suburban and rural environments is O3 in suburban and rural environments, with exposure to excess levels of these species is known to have a negative effect on human health*

Line 52-55: Statement may apply to urban centers, but it is not broadly true. $O_3$ has decreased with decreasing NOx emissions in many locations. See, for example:

Strode, S.A., J.M. Rodriguez, J.A. Logan, O.R. Cooper, J.C. Witte, L.N. Lamsal, M. Damon, B. Van Aartsen, S.D. Steenrod, and S.E. Strahan, Trends and variability in surface ozone over the United States. Journal of Geophysical Research: Atmospheres, 2015. 120(17): p. 9020-9042.

Cooper, O.R., D.D. Parrish, J. Ziemke, N.V. Balashov, M. Cupeiro, I.E. Galbally, S. Gilge, L. Horowitz, N.R. Jensen, J.-F. Lamarque, V. Naik, S. Oltmans, J. Schwab, D.T. Shindell, A.M. Thompson, V. Thouret, Y. Wang, and R.M. Zbinden, Global distribution and trends of tropospheric ozone: An observation-based review. Elem. Sci. Anth., 2014. 2: p. 29.

Cooper, O.R., R.-S. Gao, D. Tarasick, T. Leblanc, and C. Sweeney, Long-term ozone trends at rural ozone monitoring sites across the United States, 1990–2010. J. Geophys. Res., 2012. 117(D22): p. D22307.

**We have made it clear that our statement relates to urban sentence and added a sentence about more rural environments and added the suggested references:**

*In more rural environments, the opposite has been observed, with $O_3$ decreasing with decreasing $NO_x$ emissions (Cooper et al, 2012; Cooper et al., 2014; Strode et al., 2015).*

Line 65, Figure 1: The notation in the insets is quite difficult to read and will not be legible on a printed page (readability required >200% magnification on my screen).

**We have replotted and used a larger font size for the labels.**

Line 66: Figure 1 uses Google mobility data, which are qualitative at best and have shown different reductions relative to other markers, such as traffic counts, in different regions. Can the authors provide another data set, such as traffic counts, or else some statement of uncertainty in the Google mobility data? If not, a caveat should appear re: the use of these data and their uncertainty as a proxy for actual traffic counts. Google and Apple mobility data are easy to obtain but not necessarily the best measure.

**We were unable to get actual traffic flow data for cities in the UK so have used Google mobility data as a proxy for traffic. We realise this is not ideal but as the data is purely indicative we think it is reasonable to use it. We have added a caveat as the reviewer suggested (Line 70):**

*Google mobility data was used as a proxy for traffic counts as it is readily accessible, however for any quantitative analysis of the effect of reduction in traffic on pollution levels, real traffic counts or flow data would be required.*

Line 115-116: It is useful to have the description of the NOx measurements in this paper and acknowledgement of the potential for interference on Mo converters – several recent analyses of COVID impacts have not addressed this issue at all. For this last statement, however, it is apparent that many of the sites in the network are not urban. What is the bias in using Mo converter $NO_x$ for these sites? Would this really be consistent with the 15% accuracy quoted below? How much does this affect the analysis of $O_x$ later in the manuscript? As with other comments, this uncertainty should be propagated through the quantitative measures given later.

**We do not believe the use of Mo converters for the $NO_2$ measurements in this analysis is a problem. All of the sites that we analyse are urban (either traffic or urban background), so the interference will be minimal. In addition, because we are largely looking at a change in $NO_2$, any interference that is present is likely to be there in very similar amounts in the 2020 and 2015-2019 data. We do not think it is possible to give a quantitative bias for the interference across so many sites. We have added this statement:**

*In addition, as we are looking at a change in NO2, it is likely that any interference that is present will be there in very similar amounts in both the 2020 and 2015-2019 data.*

Line 133-137: What variables were used for the detrending? Was this purely a seasonal trend, or were standard meteorological and day of week variables used? How well did this fit the 5 year data? The statement regarding emissions changes lacking importance during the 5 year period does not seem consistent with the two noted changes in 2017 and 2019. Would large step changes in emissions be expected, especially if new control measures were implemented in 2019, the year before the lockdowns?

**Detrending was performed by linear regression based on monthly mean values of $NO_2$. This has been clarified in the manuscript. Each site was detrended individually before taking the mean over all sites therefore the quality of the fit varied from site to site. The mean $r^2$ value for all sites 0.60.**

**Step changes would not be expected in the time series with the introduction of Euro VI standards in 2017 or ultra-low emission zone in London 2019 and these examples are not intended to be representative of what is seen in Figure 2. Introducing the Euro emission standard will not be a step change in emissions as fleet replacement takes place over decades and the introduction of the ULEZ in London would not be seen as Figure 2 shows a mean across all of the UK. Where London $NO_2$ is singles out for analysis later in the paper, the AURN site used falls outside the ULEZ and therefore is unlikely to observe a large step change at the introduction of this policy.**

**We have removed the examples stated on line 139 as the reviewer is correct that they are not really longer-term emission changes since they both happened in the last three years – which is within our five-year analysis.**

**We have changed the sentence from line 175 to now read:**

*To deseasonalise the data, we determine the climatology based on the mean annual cycle of the previous five years (from January 1st 2015 to December 31st 2019) which is then repeated to match the length of the timeseries, subtracted from the mean to standardise the data, and then subtracted from the original timeseries to produce a timeseries of the residuals.*

Line 139, Figure 2: What is the measurement frequency or averaging period for the $NO_2$ and $O_3$ data? These appear to be approximately monthly averages? If data are heavily averaged, the figure should also show a variability? Such variability would represent both the variability in averaging to a

monthly (?) value at a given site, and the averaging of all sites. It is also not clear that averaging this collection of sites together makes sense, since there are likely large, systematic difference between sites? A relative, rather than absolute, y axis would seem to be more appropriate in this case.

**The AURN measurement frequency is hourly and measurements are averaged to monthly values at each site for figure 2. We agree with the reviewer that a relative y axis would be more appropriate and have updated the figure to incorporate this change. We have also added the 25th and 75th percentiles to show the variability between sites.**

**Figure 2 is now:**

[Figure]

Line 180-181: "Care must be taken" in the comparison due to the meteorological variability. However, there does not appear to be an effort in this paper to correct the data for meteorological variability during the lockdown period. Perhaps this will appear in a later section, but some estimate, at least, of the effect of meteorological differences on the uncertainty of the NOx and O3 changes quoted in the paper is needed.

**See above for inclusion of the standard error of the differences. We have also now plotted NO₂ against wind speed (both median across the period and the change between 2020 and 2015-2019 – figure S3) for each site and see that, whilst NO₂ concentrations do tend to be higher at lower wind speed, there is very little correlation between the magnitude of the change in NO₂ and any difference in wind speed between 2020 and 2015-2019.**

**Figure S3:**

[Figure]

Line 184-185: Figure S2 is quite difficult to read, even with magnification. The color legends are not legible, and the red lines are nearly invisible. So despite the statement here that the figure shows clear changes, it is not of sufficient quality to do so.

**We have made the red line larger and quality of the figures will improve in the final submitted version.**

Line 189-194: Were there any obvious differences in meteorology (e.g., just T for example) on the days that were above and below the long-term average?

**There is no obvious differences in meteorology on the individual days that were above and below the long term average. This is why we choose to take the bulk average of the data across the whole pre and post-lockdown period.**

Line 206 and Figure 4: What is the error bar on the 42%? See comments above – this should be shown in the figure and represent the site to site variability together with the uncertainties in the measurements and potential meteorological artefacts.

**As describe above we have added an error analysis to our data, such that any quote difference in pollution levels now comes with an error. The total of 42% has a standard error of ±9.8%.**

Line 211: Is it a t-test or a z-test? Use consistent notation and give a definition. If the NO2 data are not normally distributed (which is likely), is the test appropriate?

**We originally used a z-test in this analysis, references to T-tests in the manuscript were errors. However, we have reviewed this statistic at the recommendation of the reviewer and agree that is it not appropriate due to the data not being normally distributed. We have changed this test to a Mann-Whitney U-test which can provide the same information but is used for non-normal data. These results have been updated in the manuscript. We have also added the sentence (line 200):**

*We use this test rather than a t-test or z-test due to the large sample size and non-normal distribution of the data.*

Lines 214-220: The discussion here is somewhat unsatisfying, again due to the qualitative use of meteorology, which is rather arbitrarily applied to provide justification of a difference when one is not expected, but is not given any weight in the case where a difference is expected due to the lockdown.

**We have re-written this small section to add our error calculation and some more quantitative analysis of wind speed, which was 33% higher in the pre lockdown period in 2020 compared to 2015-2019.**

*We also find 112 sites (from a possible 128 sites, 88%) show a significant difference between NO2 measurements made in 2020 immediately prior to the lockdown to the mean of the same period from the previous five years, with urban background sites showing a -6.3±1.5 µg m$^{-3}$ change and urban traffic 9.2±1.9 µg m$^{-3}$). We attribute this difference to changes in meteorology during January and February 2020 (Figure 3), in particular wind speed, which was on average 33% higher than the average of for the previous 5 years. Finally, we find that 97 sites (75%) show a statistically significant difference between mean NO2 observations during lockdown and immediately prior to lockdown in 2020, implying there was a significant drop in NO2 across the UK as a direct result of the lockdown.*

Line 239, Figure S4: Same comment as above for Figure S2.

**We have made the red line larger and quality of the figures will improve in the final submitted version.**

Line 242: Reference is likely intended to Figure 5 rather than 4.

**This has been corrected.**

Line 247-250: The previous discussion cites only the effect of titration by NO emission in regulating the response of O3. Here photochemistry is invoked. Is the seasonal photochemistry expected to be strong in this season in the UK? For example, does O3 exceed its background values at this time of year? The discussion re: petrochemical emissions seems quite speculative in the absence of measurement or modelling.

**Certainly in April and May, seasonal photochemistry can have a large effect on $O_3$ in the UK, with frequent incidences of $O_3$ exceeding background levels. 2020 had some particularly hot and sunny periods in April and May (see discussion section). We agree that we don't have any firm evidence for oil and gas emissions being the cause of the $O_3$ decrease and we have now made it clear in the text that it is speculative:**

*This could be achieved by possible fugitive emissions from the onshore gas terminals near Aberdeen, although we have no VOC measurements to confirm this so the hypothesis is entirely speculative.*

Line 273: Same comment as above re: the use of Google mobility data as a proxy for traffic. Caveats or uncertainties required, but more reliable data sets, such as actual traffic counts, would be preferable.

**As stated above we do not have access to real traffic data and we have stated that Google activity data is purely a proxy for traffic by adding the following sentence:**

*During the lockdown period there has been around a 75% reduction in road traffic across the UK, (using Google activity data as a proxy for traffic).*

Line 288-311 and Figures 7 and S7: A 5% increase in Ox in southern UK cities is cited here. Similar to comments above, there is no statement of the uncertainty or variability in this estimate, but the authors should provide one. The changes in Ox appear to be well correlated with changes in UVA for 2020 – in other words, that the increased Ox is plausibly not attributable to reduced NOx emissions. This would also be consistent with the trend shown in Figure 7a, at least qualitatively. Is that the conclusion of this paragraph? It is not clear what is being said here. Finally, the T correlation in Figure 7b is difficult to interpret. What is the line? Is this a fit? If so, it does not appear to represent the data. The T data appear to be more related to the isoprene data in the following section, and so perhaps should be presented with that figure.

**We have changed the text to include the actual changes for the 6 cities in question and quoted errors. Yes the reviewer is correct that conclusion is that in the southern cities there is an increase in net $O_x$, therefore some of the $O_3$ increase is not solely attributable to reduced $NO_x$. We have added the following sentence at the end of the paragraph to make that clearer.**

*Therefore we conclude that in the cities in southern UK, some of the O3 increase is not solely attributable to reduced NOx, but also an increase in photochemistry related to the hot sunny weather experienced in 2020.*

**We have also removed figure 7b as we agree there is no real correlation.**

Line 321-323: Statement might be true, but it would really depend on the dominant source for VOCs. For example, if VOC were mainly from traffic emission, as NOx emissions are stated to be for this region, then one might expect similar changes in the two. Absent some statement of an inventory and major VOC sources, this statement does not appear justified.

**We agree and have removed the statement about 1,3 butadiene and 1- butene and we concentrate the discussion on benzene, for which we have emission estimate data. We now state that:**

*Indeed, in London according to the NAEI (in 2018), road transport only contributes 11% to sources of benzene, with other major sources being domestic combustion (69%), other transport (10%) and offshore oil and gas production (6%). Therefore it is not surprising that VOCs show less of a reduction during the lockdown than $NO_2$.*

Line 333, Figure 8: Why is isoprene apparently exactly zero in the historical average at Marylebone Rd. for most of the record? Is this a measurement artifact?

**We have looked at the isoprene data and see that in 2017 and 2018 it was zero for most of the time. We believe this data is not reliable so we have replotted the figure to compare just 2019 data to 2020. We also update the text to state this:**

*Figure 9 shows daytime mean (10:00 – 18:00) isoprene data and its contribution to k' at two sites in 2019 and 2020 (the only years where reliable isoprene data is available).*

Section 4.3: The literature review given here is useful, but it more likely introductory material than it is a conclusion. Suggest moving to the introduction.

**This has been moved to the introduction.**

Line 436-438: The "warning" regarding O3 is an important statement, but it is not clear that it is justified. The major influence of NOx, if I understand the conclusion of the paper correctly, is in the change in O3 titration, but not in photochemistry (at least not in the winter-spring season studied here). Thus, O3 would go to its background value in the absence of NO titration, while photochemistry would not be strongly affected. Does this scenario really constitute a "warning" that would need to be taken into account to inform emissions reduction policy? The finding is quite different from that referenced in the following sentence regarding O3 in China, where changes are clearly attributable to photochemical processes. The paper should not conflate the two regions, which are clearly quite different.

**We agree our statement is a little confusing. We have changed it so we state that the changes in the UK are largely due to reduction in $O_3$ titration with NO, rather than photochemistry and have removed the $O_3$. We have added the following sentence (line 486):**

*In China, NOx reductions have led to increases in O3 (Li et al., 2019a; Ma et al., 2016; Sun et al., 2016) and therefore air quality abatement strategies are being developed in order to offset this, largely by also controlling VOCs (Li et al., 2019b; Le et al., 2020). These changes are attributable to photochemical processes (e.g. the reduction in particles causing increased radiation and photochemistry), however our study shows that a large reduction in NOx, directly causes an increase in O3 due to a reduction in titration with NO.*

The authors address the impacts of the COVID-19 lockdown to NO2 emission reductions in the UK, and the possible implications to surface O3 levels. More specifically, they present measurements from 128 urban monitoring stations and compare the 2020 lockdown period, to the 2020 pre-lockdown period, and the same periods from 2015- 2019. They follow an approach to deseasonalise and linearly detrend the 2020 data based on the previous years to show that NO2 levels have dropped for various UK cities while O3 has increased.

**We thank the reviewer for the detailed review and hope to answer the questions below.**

Although the authors discuss the implications of meteorology to the NO2 concentration reductions these effects are not carefully taken into account. They present meteorological differences between these periods that show higher wind speeds during the lockdown and many times from different directions. A characteristic example is e.g. Cardiff that showed increased wind speeds (Fig. 3) and the highest NO2 reductions (Fig. 4) that are currently fully attributed to the lockdown. I would recommend that the authors perform a more detailed analysis of the meteorological conditions and only include the cities that had similar wind speeds, wind directions, and exclude the ones that did not.

**We thank the reviewer for the suggestion here. We agree that wind speed plays a key role in the NO$_2$ concentration, however we do not think we should exclude any cities based on meteorological conditions. We would then be setting an arbitrary limit on meteorology so we would rather keep everything in. We have added a more detailed discussion to section 3.1 on the actual changes in wind speed observed across the 6 cities, stating that Cardiff (followed closely by Bristol) do indeed show the greatest change. The other cities have a much smaller increase in wind speed compared to the previous five years. We have added the following text:**

*The six cities saw an average wind speed in 2020 of 6.5±1.2 ms-1, which was 33.5% higher than the average of the previous five years.  Since the beginning of the COVID-19 lockdown, meteorological conditions have been much more settled, with high pressure and easterly winds dominating UK weather since mid-March, especially in southern and western UK. Average wind speeds across the six cities was 4.1±0.4 ms-1, although this is still an increase of 7.5% compared to the previous 5 years. Of the cities analysed, Cardiff saw the largest increase in wind speed for 2020 compared to the previous five years (16.8%), with Bristol showing a 10% increase. The other cities all saw slight decreases in wind speed (<5%) in 2020. Typically, lower wind speed meteorological conditions are associated with higher levels of air pollution due to increased atmospheric stability and transport of*

*pollution from mainland Europe in the UK, respectively, and so care must be taken when comparing pre and post-lockdown levels of air pollution, as described in section 2.3, and comparing to the average of the previous 5 years is a better measure of the changes.*

**We have also plotted the change in $NO_2$ against the change in wind speed for each site, along with median $NO_2$ against median wind speed for the lockdown period in 2020 and 2015-2019 - this figure is now in the SI (figure S3 – see above in reviewer 1 response). It shows that, whilst $NO_2$ concentrations do tend to be higher at lower wind speed, there is actually very little correlation between a change in wind speed between the two periods and the change in $NO_2$. We have added the following sentence (line 254):**

*We see that, whilst NO2 concentrations do tend to be higher at lower wind speed (Figure S3(a)), there is very little correlation between the observed change in NO2 between 2020 and the previous five years and any change in wind speed (Figure S3(b)).*

When moving to the O3 trends things become more challenging since O3 is strongly affected by meteorology as well as NOx and VOC emissions (the lifetimes of the latter also affected by meteorology). Although O3 formation is complicated the authors seem to oversimplify it and often promote a link between O3 formation and NOx reductions that is not supported at all by the observations. On the contrary, observations promote differences in the UV levels that could drastically increase O3 compared to previous years.

**We do not entirely agree with this comment. We do see that there is a clear correlation between observed $NO_2$ and observed $O_3$ across the sites. We have added a new figure (6) to show this. It shows the change in $O_3$ concentration between 2020 and the previous 5 years, plotted against the change in $NO_2$ concentration and demonstrates a correlation. (b) shows the median $O_3$ concentration for the lockdown period at each site plotted against the median $NO_2$ concentration. This also shows a clear anti-correlation between $NO_2$ and $O_3$. We have added some text describing this figure (line 346):**

*Figure 6 shows the relationship between $NO_2$ and $O_3$ during the lockdown period, across all the AURN sites we examined. There is a clear anti-correlation between median NO2 and median O3 for all data, with data from 2020 tending towards lower $NO_2$ and high $O_3$ (Figure 6(a)). We also see that there is a correlation between the change in $NO_2$ and the change in $O_3$ between 2020 and the previous five years (Figure 6(b)).*

**Figure 6:**

[Figure]

Overall, the manuscript would be suited for ACP after (1) a careful exclusion of cities that had different meteorological conditions from 2020 to 2015-2019, and (2) more honest and precise conclusions regarding the increased O3 levels.

**We thank the reviewer for this comment and try to answer their concerns below.**

*Specific comments:*

Page 1, lines 17-18: ". . . suggesting the majority of this change can be attributed to photochemical repartitioning due to the reduction in NOx.". The authors did not quantify the effect of meteorology and NOx reductions to be able to conclude this. Please rephrase.

**We believe we have now shown that the majority of O₃ change is due to NOₓ reductions.**

Page 1, line 21-22: Can the authors make this statement without looking in more detail the meteorological differences between the studied years?

**We are just reporting what is observed so we feel this statement is still valid.**

Page 1, line 37-39: Where is the remaining 16% NOx coming from? Please provide in parenthesis the variability as ± XX%. Also, there is no contribution of biomass burning to NOx which especially in the wintertime could play a role.

**We have added the contribution from other sources to the section. Note we have updated the figures so they are from the 2018 inventory, which was not available at the time of the original manuscript submission. We also now quote the variability (standard deviation) of the NAEI contributions across the 6 cities that we have used in the other analysis (London, Bristol, Cardiff, Newcastle, Glasgow, Belfast). The section now reads:**

*In 2018 the road transport sector accounted for 37% of UK NOx (sum of NO and NO2), the largest emission from a single sector,  followed by energy industries (21%), non-road transport (mainly rail and aviation) (15%), manufacturing industries and construction (10%) and domestic combustion (9%) (https://www.gov.uk/government/statistical-data-sets/env01-emissions-of-air-pollutants). In major cities, the contribution from road transport is typically much higher. On average across six cities in the UK (London, Bristol, Cardiff, Newcastle, Glasgow and Belfast)e.g. 47±6 % comes from road transport, , with 17±5 % from domestic combustion, 15±6 % from non-road transport sources (mainly rail), and 14±10 % from energy industries and 6±2 % from industrial combustion.*

Section 2.3, line 133-134: ". . . we first linearly detrend and deseasonalise NO2 data at each AURN site based on the climatology of the previous five years". Please, elaborate more and show characteristic examples of data before and after deseasonalising in the main text or SI. It was not clear to me what is shown in Fig. 2 and I had to spend a long time before understanding the de-seasonalisation approach (not 100% sure I still do). This is an important step for this study and is only very briefly discussed. This also includes the associated uncertainties.

**We agree our method is not totally clear. To de-seasonalise the data, we determine the climatology based on the mean annual cycle of the previous five years (from January 1[st] 2015 to December 31[st] 2019) which is then repeated to match the length of the time series, subtracted from the mean to standardise the data, and then subtracted from the original time series to produce a time series of the residuals. We have updated the description (line 177).**

*To deseasonalise the data, we determine the climatology based on the mean annual cycle of the previous five years (from January 1st 2015 to December 31st 2019) which is then repeated to match the length of the time series, subtracted from the mean to standardise the data, and then subtracted from the original time series to produce a time series of the residuals.*

Section 2.3, line 147: It is surprising to me that this sudden drop in January-February is suggested to be only due to emerging crises in nearby European cities. The authors later discuss that meteorology is significantly different for these months compared to March-May but still not that drastically different compared to the same months from previous years (Figure 3). I consider it important to understand where this drastic drop in concentration before the lockdown even started, is coming from. This rapid change not related to the pandemic is strong proof that this approach may not work since the needed weight to meteorology or other factors is not accounted for. If differences in meteorology between the 2015-2019 pre-lockdown, and the 2020 pre-lockdown are the reason for this drop in NO2 concentrations (which I suppose mostly is as also discussed in section 3.1) then similar differences during the lockdown (e.g. Cardiff) could play a crucial role in reduced NO2 concentrations.

**We do believe the drop in $NO_2$ concentrations in early 2020 is due to much larger wind speeds early in 2020 compared to the previous five years. however as now stated in the text (section 3.1 - see above) wind speeds for the lockdown period were not as much larger in 2020 compared to previous years (only 7.5%). Our new figure S3 (see above) also shows that that is very little correlation between the change in wind speed and the change in $NO_2$ between 2020 and the previous five years. Therefore we do believe that our approach is valid, even without a quantitative assessment of the meteorology.**

Section 3.2, line 186-189: The authors already showed how strong influence meteorology could have on the trends based on the pre-lockdown period. If a comparison for the different years was made it should be followed (and weighted) by a comparison of wind direction, wind speeds. For example, Cardiff that has higher wind speeds in 2020 compared to other years (Figure 3) has the highest drop in NO2 which is not due to the lockdown alone. Also, it would be great to see the bars in Figure 4 colored based on the concentrations observed at each site, and with error bars.

**Figure S3 shows that there is very little correlation between the change in wind speed and the change in $NO_2$ between 2020 and the previous five years, therefore we do not think colouring the bars in figures 4 would add anything.**

Line 209: Is this the mean of all 4 years from 2015-2019? I wonder whether it would make more sense to compare only to 2019. More detailed sensitivity analysis and discussion will improve the presented results here and show whether uncertainties are higher than the observed trends.

**We have now added errors to all the quoted concentrations and differences (see response to reviewer 1). We are confident that comparing 2020 to the average of the previous five years is the most appropriate for our analysis.**

Line 227-230: What is the contribution of biomass burning to NOx? The increase in the later hours promotes the possible effects of residential heating. Please discuss the contribution of other emission sources further in the main text.

**Domestic combustion makes up around 20% of $NO_x$ emissions in UK cities. We agree this could be the reason for the change in diurnal cycles during 2020, as it is likely not to have changed much during the lockdown. NO other source sectors make a significant contribution We have added the following sentence to the text:**

*A reason for these observed diurnal cycles could be domestic combustion, which typically makes up around 17% of urban NOx emissions (compared to 47% for road transport and 15% for other transport e.g. rail). We do not expect domestic combustion to have changed much during the lockdown, therefore its contribution to the total (and the diurnal cycle) will be greater.*

Section 3.3, line 250: Photochemistry is a key driver for O3 production. However, the authors here don't address the possible effect of yearly variations in photochemistry. Comparing j-NO2 for the different years during these periods would be essential to answering this.

**We do not have measurements of j-$NO_2$, however we do discuss changes in UV radiation in the discussion section. We do not believe radiation is a factor here because, as stated later, UV radiation increase in 2020 if anything and this observation is a large decrease.**

Line 288-311: Aren't the authors suggesting here that the increased O3 is mostly due to meteorology? Please emphasize this more and de-emphasize the O3 increase due to NOx reductions since there is no trend to support this.

**We are saying that the majority of the change in $O_3$ at these urban sites is due to a reduction of $NO_x$ and thus reduction in the titration of $O_3$ with NO. We have added text to this section and a figure showing the anti-correlation between the two species (see answer to a previous comment). But we are saying that meteorology does have a small effect in the southern cities, which show a small increase in observed $O_x$ ($O_3$ + $NO_2$). We hope this section is now clearer following our changes.**

Line 313-331: Various sources of VOCs and oxygenated VOCs are not discussed here, e.g. biomass burning, volatile chemical products, industry, that can play a crucial role in determining the total VOCs and total reactivity, and therefore understanding O3 formation. Presented here is not the total VOCs or total reactivity since the discussed VOCs are predominantly related to combustion/traffic emissions. In general, please emphasize more the variability of VOC emissions and that to understand O3 formation NOx and VOC emissions are equally important.

**We have added some text to this section describing more fully the contribution of other sources to VOCs.**

*Indeed, in London according to the NAEI (in 2018), road transport only contributes 11% to sources of benzene, with other major sources being domestic combustion (69%), other transport (11%) and offshore oil and gas production (6%). Therefore it is not surprising that VOCs show less of a reduction during the lockdown than NO2.*

**We also added the following text later on in the section (line 399) as a caveat to our very basic OH reactivity analysis:**

and it is unlikely that the measurements made at the AURN sites cover all VOCs that contribute to OH reactivity (e.g. few oxygenated compounds or larger VOCs are measured).

Line 341: Nothing is clear based on the presented results. The authors have no proof that O3 increased due to changes in NOx or changes in meteorology or VOCs. Please rephrase.

**We agree the wording is too string here. We have rewritten the final sentences to read:**

*Further detailed chemical modelling studies, beyond the scope of this study, are required to assess in detail the chemistry behind O3 formation and how this has been affected by the lockdown, however it is clear we observe that O3 has increased across the UK and see a clear anti-correlation with a decrease in NO2 across the sites. due to the reduction in NOx, We also see with an increase in total Ox at Urban Background sites in the South of England, . This is likely due to increased radiation and biogenically emitted VOCs compared to previous years, things that are unlikely to be linked to the COVID-19 lockdown.*

Figure 9 is since January although the lockdown was not in effect. How many exceedances happen during the pre-lockdown period? Please separate the two periods and further discuss them if necessary.

**We have changed our analysis (and updated the figure) so we only report the lockdown period. It does not change the conclusions.**

Line 427: The increase in Ox can be due to differences in UV levels that will increase OH and O3 levels as mentioned by the authors in the main text. Please rephrase.

**We believe we already state this further down the section (Line 495):**

*Whilst anthropogenic VOCs are slightly decreased during the lockdown, we find some evidence that suggests that biogenic VOCs such as isoprene are higher due to warmer temperatures and higher UV levels across southern UK in 2020 compared to previous years; we find no evidence to suggest that higher UV levels were due to cleaner skies related to air pollution changes due to the lockdown.*

Line 436-438: This is a stretch when there is no quantification of the factors affecting O3 formation. Please rephrase.

**We have reworded the sentence so it now reads:**

*If we are to take the COVID-19 lockdown as an analogue of how air quality will respond to future reductions in emissions from vehicles (e.g. over the next 10-20 years), then observations show that there could be a corresponding increase in O3 which should be considered in any air quality abatement strategy.*

Line 441-443: Strong wording. Please rephrase.

**We have reworded the sentence so it now reads:**

*In addition, a warming climate may lead to  increased emissions of biogenic VOCs, further adding to the O3 burden*

*Technical comments:*

Page 2, line 49: O3 is the main pollutant for urban pollution too. Please rephrase.

**We have added the word urban to the O₃ part of the sentence.**

Page 2, line 78: Change "has" to "could have".

**Done**

Page 3, line 96: correct to "levels are".

**Done**

Page 4, line 121: delete "and". Also, an error is provided for the PM2.5 measurements but there is no mention of the type of instrumentation used. Since PM2.5 is not used at all in this study the authors could completely skip this.

**We have removed the part about the PM₂.₅ error.**

Line 202: correct to "increase".

**Done**

Line 213: Do you mean "Observed variations in O3 will also reflect changes in precursor VOC emissions"? Even then, how would that happen? Please rephrase.

**We think the reviewer means line 313 for this and agree the sentence is not a good one. We rephrase to:**

*Observed variations in O3 may also be affected by will also reflect changes in precursor VOCs*

Line 240: correct "O3".

**Done**

Line 278: delete "however".

**Done**

*Figures comments:*

Please improve the quality of the figures in the main text and supplement. Also, include uncertainties/error bars to the figures.

**This will happen in the final version**

Figure 2: Could the authors add the 25th and 75th percentile? Also, could the authors present the results for urban and background environments in the SI for cases where this approach works and cases where this approach is more challenging?

**I've changed this figure to address a comment from review 1. The figure now shows the mean relative change of NO2 and O3. The 25th & 75th percentile have also been added as a shaded area around the mean and the caption has been adjust to match.**

Figure 6: x-axis label is missing.

**Now added.**

[revised manuscript text omitted]

---

## Author Response (AR2)

The final editorial comment was "The figures need final checks regarding layout and readability". We have done this and uploaded the figures as separate, high resolution 'png' files in accordance with submission instructions.

We also noticed three minor errors in the text that needed correcting.

**On Line 404 the sentence should read:**

*Biogenic emissions of isoprene, originating from a variety of trees and shrubs, are driven in part by temperature and so it is perhaps not surprising that isoprene levels at the London sites were higher in 2020 compared to 2019 due to the fact that temperature was approximately 2$^o$C higher in 2020 compared to 2019 for the lockdown period..*

**On line 432 there was a very slight error in the number of NO$_2$ exceedances. This is corrected and the text now reads:**

*At roadside sites, exceedances dropped consistently from 275 in 2016 to 13 in 2019, with 9 of these 13 at a site in Wandsworth (Putney High Street).*

**On line 449 was also a very slight error in the number of O$_3$ exceedances. This is corrected and the text now reads:**

[revised manuscript text omitted]